# Convergence and Connectivity: Asymptotic Dynamics of Multi-Agent Q-Learning in Random Networks

## Abstract

Beyond specific settings, many multi-agent learning algorithms fail to converge to an equilibrium solution, instead displaying complex, non-stationary behaviours such as recurrent or chaotic orbits. In fact, recent literature suggests that such complex behaviours are likely to occur when the number of agents increases. In this paper, we study Q-learning dynamics in network polymatrix normal-form games where the network structure is drawn from classical random graph models. In particular, we focus on the Erdős-Rényi model, which is used to analyze connectivity in distributed systems, and the Stochastic Block model, which generalizes the above by accounting for community structures that naturally arise in multi-agent systems. In each setting, we establish sufficient conditions under which the agents' joint strategies converge to a unique equilibrium. We investigate how this condition depends on the exploration rates, payoff matrices and, crucially, the probabilities of interaction between network agents. We validate our theoretical findings through numerical simulations and demonstrate that convergence can be reliably achieved in many-agent systems, provided interactions in the network are controlled.

## 1 Introduction

The development of algorithms for multi-agent learning has produced a number of successes in recent years, solving challenging problems in load-balancing (Krichene et al., 2014; Southwell et al., 2012), energy management (Maddouri et al., 2018) and game playing (Moravčík et al., 2017; Brown & Sandholm, 2019; Samvelyan et al., 2019; Perolat et al., 2022). Also thanks to these successes, the game-theoretic foundations of learning in the face of many agents remain a thriving area of research. As the number of agents grows in these systems, it is critical to understand if their learning algorithms are guaranteed to converge to an equilibrium solution, such as a Nash Equilibrium.

Unfortunately, previous work suggests that non-convergent behaviour becomes the norm as the number of players increases. A strong example is Milionis et al. (2023), which introduced a game in which no independent learning dynamics converges to a Nash Equilibrium. Further studies have shown that chaotic dynamics may occur even in two-player finite-action games (Sato et al., 2002; Galla & Farmer, 2013). Crucially, Sanders et al. (2018) found that, as the number of players grows, non-convergent behaviour becomes the norm.

Whilst at first glance these results suggest an insurmountable barrier towards strong convergence guarantees, a key missing factor is an in-depth analysis of the interactions between agents. In particular, both Sanders et al. (2018) and Hussain et al. (2023) assume that the payoff to any given agent is dependent on *all* other agents in the environment. This assumption rarely holds in practice. Rather, agents often interact through an underlying network that may represent communication constraints or spatial proximity. A practitioner or system designer has a certain degree of control over the network structure, e.g., in a robotic swarm, a practitioner can influence the network by adjusting the communication range of the robots, while in a sensor network, this can be done by controlling the density of the deployed sensors. This leads us to study the following research question:

*How does network structure affect the convergence of learning as the number of agents increases?*

Answering this question represents a key step towards guaranteeing the feasibility of learning with many agents, so long as the network structure can be controlled. Indeed, a number of works have examined learning in network games, uncovering the relationship between the network structure and properties of the equilibrium as well as designing algorithms that converge to an equilibrium (Melo, 2018; Parise & Ozdaglar, 2019; Melo, 2021; Shokri & Kebriaei, 2020). However, many of these algorithms have unrealistic requirements, such as full knowledge of the agents' payoff functions or their gradients. Our goal instead is to consider a widely applied reinforcement learning algorithm – *Q-Learning* – which requires only evaluations of the payoff functions, and study how the parameters of the algorithm and the structure of the underlying network can be leveraged to yield convergence guarantees, as well as how such guarantees scale with the number of agents.

**Model.** We study agents who update their strategies via the Q-Learning dynamics (Sato & Crutchfield, 2003), a continuous-time counterpart to the well-studied Q-Learning algorithm (Watkins & Dayan, 1992; Sutton & Barto, 2018). We focus our study on *network polymatrix games* (Janovskaja, 1968) in which agents select from a finite set of actions and interact with their neighbours on an underlying network. We study networks which are drawn from *random graph* models: the *Erdős-Rényi* model (Erdös & Rényi, 2006) and the *Stochastic Block model* (Holland et al., 1983). The Erdős-Rényi model is widely used to study communication in distributed systems Lei et al. (2020); Prakash et al. (2020). The Stochastic Block model is a natural extension of Erdős-Rényi to model community structures which naturally arise in some distributed systems (Yun & Proutière, 2019).

**Contributions.** Our main contribution is to characterise the convergence properties of Q-Learning, depending on the exploration rates of the agents, the game payoffs, and the expected degree of the network. Specifically, our results provide sufficient conditions for the Q-learning dynamics to converge to an equilibrium with high probability as the number of agents $N$ increases, given that the network is drawn from one of the two above-mentioned models. We demonstrate that in low-degree networks, Q-learning converges even with low exploration rates and with a large number of agents. In fact, our experiments show convergence even with 200 agents in low-degree networks, whereas Sanders et al. (2018) reported failure to converge with low exploration rates for as few as five agents.

Our work establishes an explicit relationship between the convergence of Q-Learning Dynamics and the expected node degree in the network, thus ensuring the feasibility of learning in many-agent games, so long as the expected node degree is controlled. Our results further ensure the uniqueness of the equilibrium, meaning that Q-learning converges to a single solution regardless of the initial conditions. To the best of our knowledge, this is the first work to study the asymptotic behaviour of learning dynamics in the context of network polymatrix games with random graph models, and to derive the relation between convergence and the expected node degree in the network.

## 1.1 RELATED WORK

Our work focuses on independent online learning in network polymatrix games. Network games are well-studied in the setting of zero-sum networks (Cai et al., 2016), which model strictly competitive systems. In such cases, it is known that the continuous-time counterparts of popular algorithms such as Fictitious Play (Ewerhart & Valkanova, 2020) and Q-Learning (Leonardos et al., 2024) converge to an equilibrium. By contrast, it was shown in Bailey & Piliouras (2019) that the *Replicator Dynamics* (Maynard Smith, 1974), a continuous-time model of the Multiplicative Weights Update algorithm (Arora et al., 2012) does not converge to the Nash Equilibrium. Similarly, Shapley (1964) showed the non-convergence of Fictitious Play in a two-person non-zero-sum game. Indeed, Hart & Mas-Colell (2003) showed that *no* learning dynamic that is uncoupled, in that the strategy of a player is independent of the payoff functions of other players, can converge to a Nash Equilibrium in all normal-form games. Milionis et al. (2023) extended this result to show the impossibility of convergence of a wide class of learning dynamics to approximate Nash-Equilibria in normal-form games.

Network games have also been studied to understand the properties, in particular the uniqueness, of the equilibrium (Bramoullé et al., 2014; Parise & Ozdaglar, 2019; Melo, 2021). In many cases, the literature appeals to the study of *monotone games* (Paccagnan et al., 2018) which subsumes zero-sum network games (Akin & Losert, 1984). The formalism of monotone games has been applied to design algorithms that provably converge to Nash Equilibria. However, many of these algorithms

require that agents have full access to their payoff function (Parise & Ozdaglar, 2019) or its gradient (Mertikopoulos & Zhou, 2019). Monotone games also share strong links with the idea of *payoff perturbations* (Facchinei & Pang, 2004) in which a strongly convex penalty is imposed to agents' payoff functions to stabilise learning (Abe et al., 2024; Sokota et al., 2023; Liu et al., 2023).

Our work departs from the above by considering *Q-Learning Dynamics*, a foundational exploration-exploitation model central to reinforcement learning (Albrecht et al., 2024a; Tuyls, 2023). Q-Learning Dynamics is also related to the replicator dynamics (Bloembergen et al., 2015) and to Follow-the-Regularised-Leader (Abe et al., 2022). Outside of specialised classes of games, Q-Learning Dynamics has been shown to exhibit chaotic orbits (Sato et al., 2002), a phenomenon which becomes more prevalent as the number of players increases (Sanders et al., 2018). Similar to our work, Hussain et al. (2024) study the convergence of Q-Learning in network polymatrix games. However, their work considered deterministic graphs with specific structures. We instead consider Q-Learning in a stochastic setting, where stochasticity arises from the random network. In doing so, we derive a direct relationship between the expected node degree and the convergence of Q-Learning Dynamics.

To achieve our result, we appeal to the framework of *random networks*, which are often used to model decentralised systems. Many works in multi-agent learning and online optimisation have considered this setting, for example to determine the existence of pure Nash Equilibria (Daskalakis et al., 2011), to study the performance of distributed online algorithms (Lei et al., 2020), or to examine the emergence of cooperative behaviors (Dall'Asta et al., 2012). A parallel step in the study of network games was the introduction of *graphon games* in Parise & Ozdaglar (2023); Carmona et al. (2022). The authors consider network games in the limit of uncountably infinite agents and generalise the Erdős-Rényi and Stochastic Block models. This also extends mean-field games (Laurière et al., 2022; Hu et al., 2019) by introducing heterogeneity amongst players through their edge connections. Subsequent works on graphon games largely focus on the analysis of equilibria (Caines & Huang, 2021; Aurell et al., 2022) or design learning algorithms that converge in time-average to an equilibrium (Cui & Koeppl, 2022; Zhang et al., 2023; 2024). By contrast, our goal is to understand the *last-iterate* behaviour of Q-Learning Dynamics and establish probabilistic bounds to guarantee convergence in games with finite players. In particular, we study how the probability of edge connections in the network curtails non-convergent dynamics as the number of players increases. Thus, the restriction to uncountably infinite players is not suitable for our purposes, although the analysis of non-convergent learning algorithms in graphon games is an interesting direction for future work.

## 2 BACKGROUND

**Game Model.** In this work, we consider *network polymatrix games* (Janovskaja, 1968; Cai et al., 2016), which are defined as tuples $\mathcal{G} = (\mathcal{N}, \mathcal{E}, (\mathcal{A}_k)_{k \in \mathcal{N}}, (A^{kl}, A^{lk})_{(k,l) \in \mathcal{E}})$, and where $\mathcal{N} = \{1, 2, \ldots, N\}$ denotes a set of $N$ agents, indexed by $k$. The interactions between agents is modelled by a set $\mathcal{E} \subset \mathcal{N} \times \mathcal{N}$ of edges that defines an *undirected network*. An alternative formulation of the underlying network is through an *adjacency matrix* $G \in \mathbb{R}^{N \times N}$ in which $[G]_{kl} = 1$ if $(k,l) \in \mathcal{E}$, and $[G]_{kl} = 0$ otherwise. The *degree* of an agent $k \in \mathcal{N}$ is the number of edges in $\mathcal{E}$ that include $k$.

At each round, each agent $k$ selects an action $i \in \mathcal{A}_k$, where $\mathcal{A}_k$ is a finite set of $n_k$ actions. We denote the *strategy* of an agent as the probability distribution over their actions. Next, we define the *joint strategy* across all agents as the concatenation of all individual strategies $\mathbf{x} = (\mathbf{x}_1, \ldots, \mathbf{x}_N)$ and apply the shorthand $\mathbf{x}_{-k}$ to denote the strategies of all agents other than $k$.

The goal of each agent is to maximise a utility function $u_k$. In a network polymatrix game $\mathcal{G}$, each edge is associated with the *payoff matrices* $(A^{kl}, A^{lk})$, i.e., the payoff for each agent $k$ takes the form

$$u_k(\mathbf{x}_k, \mathbf{x}_{-k}) = \sum_{l:(k,l) \in \mathcal{E}} \mathbf{x}_k^\top A^{kl} \mathbf{x}_l.$$

**Solution Concepts.** In this work, we focus on two widely-studied solution concepts: the *Nash Equilibrium* (Nash, 1950) and *Quantal Response Equilibrium* (McKelvey & Palfrey, 1995). To define these concepts we first define the *reward* to agent $k \in \mathcal{N}$ for playing action $i \in \mathcal{A}_k$ as $r_{ki}(\mathbf{x}_{-k}) = \sum_{l:(k,l) \in \mathcal{E}} \sum_{j \in \mathcal{A}_l} [A^{kl}]_{ij} x_{lj}$.

**Definition 1** (Nash Equilibrium). A joint strategy $\mathbf{x}^* \in \Delta(\mathcal{A})$ is a Nash Equilibrium if

$$\mathbf{x}_k^* \in \underset{\mathbf{x}_k \in \Delta(\mathcal{A}_k)}{\arg\max} \, u_k(\mathbf{x}_k, \mathbf{x}_{-k}^*) \text{ for all } k \in \mathcal{N}.$$

**Definition 2** (Quantal Response Equilibrium (QRE)). Let $T_1, \ldots, T_N \geq 0$. Then, a joint strategy $\mathbf{x}^* \in \Delta(\mathcal{A})$ is a Quantal Response Equilibrium if

$$\mathbf{x}_k^* = \frac{\exp\left(r_{ki}(\mathbf{x}_{-k}^*)/T_k\right)}{\sum_{j \in \mathcal{A}_k} \exp\left(r_{kj}(\mathbf{x}_{-k}^*)/T_k\right)} \text{ for all } k \in \mathcal{N}.$$

The QRE is a natural extension of the Nash Equilibrium that allows agents to play suboptimal actions with non-zero probability. This is crucial in online learning, where agents must *explore* their strategy space. The parameter $T_k$ is therefore known as the *exploration rate* of agent $k$. Notice that, by taking the limit $T_k \to 0$ for all $k$, the QRE converges to the Nash Equilibrium (McKelvey & Palfrey, 1995).

**Q-Learning Dynamics.** We now describe the Q-Learning algorithm (Watkins & Dayan, 1992; Sutton & Barto, 2018) which aims to learn an optimal action-value function $Q : \mathcal{A} \to \mathbb{R}$ that captures the expected reward of taking a given action. We consider the multi-agent extension of Q-Learning (Schwartz, 2014; Albrecht et al., 2024b) in which each agent $k$ maintains an individual Q-value estimate $Q_k : \mathcal{A}_k \to \mathbb{R}$. These are updated at each round $t$ via the update

$$Q_{ki}(t+1) = (1 - \alpha_k)Q_{ki}(t) + \alpha_k r_{ki}(\mathbf{x}_{-k}(t)), \tag{1}$$

where $\alpha_k \in (0, 1)$ is the learning rate of agent $k$.

In effect, $Q_{ki}$ gives a discounted history of the rewards received when action $i$ is played, with $1 - \alpha_k$ as the discount factor. Note that Q-values are updated by the rewards $r_{ki}(\mathbf{x}_{-k})$ that depend on the strategies of all other agents at time $t$. So, the reward for a single action can vary between rounds. This non-stationarity often leads to chaotic dynamics in multi-agent learning.

Given the Q-values, each agent plays their actions according to the *Boltzmann* distribution:

$$x_{ki}(t+1) = \frac{\exp\left(Q_{ki}(t+1)/T_k\right)}{\sum_{j \in \mathcal{A}_k} \exp\left(Q_{kj}(t+1)/T_k\right)}, \tag{2}$$

where $T_k \in (0, \infty)$ is the exploration rate of agent $k$. Let $\langle \cdot, \cdot \rangle$ denotes the scalar product.

Tuyls et al. (2006) and Sato & Crutchfield (2003) showed that a continuous-time approximation of the Q-Learning algorithm is given by a variation of the replicator dynamics (Maynard Smith, 1974; Hofbauer & Sigmund, 2003) that is called the *Q-Learning Dynamics*

$$\frac{\dot{x}_{ki}}{x_{ki}} = r_{ki}(\mathbf{x}_{-k}) - \langle \mathbf{x}_k, r_k(\mathbf{x}) \rangle + T_k \sum_{j \in \mathcal{A}_k} x_{kj} \ln \frac{x_{kj}}{x_{ki}}. \tag{QLD}$$

Leonardos et al. (2024) proved the fixed points of (QLD) coincide with the QRE of the game.

**Additional Notation.** Given a square matrix $A \in \mathbb{R}^{N \times N}$, denote its *spectral norm* as $\|A\|_2 = \sup_{x \in \mathbb{R}^N : \|\mathbf{x}\|_2 = 1} \|Ax\|_2$. If $A$ is symmetric, all its eigenvalues $\lambda_1, \ldots, \lambda_N$ are real, and the spectral norm agrees with the *spectral radius*, which is defined as $\rho(G) = \max\{\lambda_1, \ldots, \lambda_N\}$.

## 3 CONVERGENCE OF Q-LEARNING IN GRAPHS

In this section, we show that the convergence of (QLD) is closely related to the structure of the underlying graph. First, we present the problem setup: we specify the payoffs by the *intensity of identical interests* framework (Hussain et al., 2024) and specialize to a certain class of network polymatrix games, which satisfy Assumption 1. Second, we establish in Lemma 1 a sufficient condition on the exploration rates $T_k$ such that (QLD) converges to the *unique* QRE. We adapt this result to the random network setup using the Erdős-Rényi and Stochastic Block models. In both cases, we establish connections between (QLD) and the expected degree of a node in the network.

**Definition 3** (Intensity of Identical Interests). Let $\mathcal{G} = (\mathcal{N}, \mathcal{E}, (\mathcal{A}_k)_{k \in \mathcal{N}}, (A^{kl}, A^{lk})_{(k,l) \in \mathcal{E}})$ be a network polymatrix game. Then the intensity of identical interests $\delta_I$ of $\mathcal{G}$ is given by

$$\delta_I = \max_{(k,l) \in \mathcal{E}} \|A^{kl} + (A^{lk})^\top\|_2. \tag{3}$$

The intensity of identical interests $\delta_I > 0$ measures the similarity between the payoffs of connected agents across all edges. A canonical example is the *pairwise zero-sum game* in which $A^{kl} = -(A^{lk})^\top$ for all $(k, l) \in \mathcal{E}$. In this case, the intensity of identical interests is zero.

To prove our main result, we make the following assumption on the network polymatrix games.

**Assumption 1.** Each edge is assigned the same bimatrix game, i.e., $(A^{kl}, A^{lk}) = (A, B) \, \forall (k, l) \in \mathcal{E}$.

Note that Assumption 1 does not require $A^{kl} = A^{lk}$, rather it requires that each edge is associated with the same *pair* of matrices $(A, B)$. No assumption is made about which agent receives matrix A and which receives matrix B. This is reflected in our experiments (Section 4) in which payoff matrices are randomly assigned to agents on each edge. This aspect of the setting is discussed in detail in Appendix D.

Assumption 1 is well motivated in the literature as it analyses the case where agents are engaged in the same interaction with multiple opponents, a format that is commonly used to study the emergence of cooperation (Zhang et al., 2014; Mukhopadhyay & Chakraborty, 2020) and congestion (Szabó & Szolnoki, 2015) in multi-agent systems. Indeed, the assumption of shared payoffs has led to a number of successes in studying large scale systems with many agents (Perrin et al., 2020; Wu et al., 2024; Pérolat et al., 2022; Parise & Ozdaglar, 2023; Hu et al., 2019). We adopt this assumption, but also empirically validate our claims outside of this framework through the *Conflict* Network game (Ewerhart & Valkanova, 2020), in which payoff matrices vary across edges; see Section 4 for details.

Having specified our setting, we next determine a sufficient condition for the convergence of (QLD) in terms of the properties of the adjacency matrix $G$. All proofs are deferred to Appendix B and C.

**Lemma 1.** Let $\mathcal{G} = (\mathcal{N}, \mathcal{E}, (\mathcal{A})_k, (A, B)_{(k,l) \in \mathcal{E}})$ be a network polymatrix game that satisfies Assumption 1. Also, let $G$ be the adjacency matrix associated with the edge-set $\mathcal{E}$. If

$$T_k > \delta_I \rho(G), \text{ for all } k \in \mathcal{N} \tag{4}$$

the QRE $\mathbf{x}^*$ of the game $\mathcal{G}$ is unique. Further, $\mathbf{x}^*$ is globally asymptotically stable under (QLD).

This result has two components: (i) the QRE is unique, and (ii) it is asymptotically stable with respect to (QLD). Together, these statements ensure that a unique equilibrium solution is learned.

**Discussion**  Recall from Section 2 that lower exploration rates are preferred, so that the equilibrium solution remains close to a Nash Equilibrium. Lemma 1 provides two quantities, $\delta_I$ and $\rho(G)$ which can be chosen appropriately to guarantee the convergence of (QLD). A common approach to address this is to fix the game to a specific class. As an example, restricting to pairwise zero-sum games, where $\delta_I = 0$, ensures that (QLD) converges as long as $T_k > 0$ for all $k \in \mathcal{N}$, regardless of $\rho(G)$. Our approach, in contrast, focuses on controlling the spectral radius, $\rho(G)$, while treating $\delta_I$ to be fixed a-priori. To achieve this, we explore how network sparsity—a tunable parameter in decentralised systems such as robotic swarms or sensor networks—impacts the convergence of (QLD). Our analysis turns to *random networks*, where network sparsity is directly parameterised through the probability of edge connections. By deriving probabilistic upper bounds on the (random) spectral radius term $\rho(G)$ in terms of these parameters, we can directly link the network's sparsity to the convergence or (QLD), providing robust convergence guarantees that hold for broad classes of networks.

We specify the network structure by the parameters of the model from which it is generated. In the Erdős-Rényi model, this is the probability $p$ that a pair of nodes is connected by an edge. In the Stochastic Block model, it is the set of within-community edge probabilities $p_1, \ldots, p_C$ and between-community edge probability $q$. We stress that our theoretical bounds are not limited to these specific models: they apply to any random network model with independent entries (see Lemma 9).

**Erdős-Rényi.**  We begin with the Erdős-Rényi (ER) model (Erdős & Rényi, 2006), in which the graph is generated by independently sampling each edge with probability $p \in (0, 1)$. The adjacency

matrix $G$ is then a random matrix with entries that are Bernoulli distributed with parameter $p$. Let the degree of a node $k$ be the number of edges $(k, l) \in \mathcal{E}$ that contain $k$. The expected node degree in a network drawn from the ER model is therefore $p(N - 1)$.

**Stochastic Block Model.** Networks are drawn from the Stochastic Block (SB) model (Holland et al., 1983) by partitioning the nodes into $C$ communities. The probability of an edge between nodes $k$ and $l$ that are in the same community $c$ is given by $p_c \in (0, 1)$, whereas the probability of an edge between nodes in different communities is $q \in (0, 1)$. The adjacency matrix $G$ is then generated by sampling each edge independently according to the Bernoulli distribution with probabilities $p_c$ and $q$. The expected node degree in community $c$ is $p_c(N_c - 1) + q(N - N_c)$, where $N_c$ is the number of nodes in community $c$. Note that the higher $p_c$ and $q$, the higher the expected node degree, and further that the ER model is a particular case of the SB model for a single community, i.e., $N_c = N$.

We first place probabilistic bounds on the spectral radius of the adjacency matrices in the ER and SB models, which we then use in Theorems 1 and 2 to obtain the main convergence results. For simplicity, in the SB model case, the following results assume all communities are of equal size, specifically $N_c = N/C$ for all $c$. However, the proof stays valid when the communities' sizes vary.

**Lemma 2.** Consider the symmetric adjacency matrix $G \in \mathbb{R}^{N \times N}$ of a network drawn from the ER model, i.e., $G$ has a zero main-diagonal and the off-diagonal entries $\{[G]_{kl} \sim \texttt{Bernoulli}(p) : 1 \leq k < l \leq N\}$ are independent, where $p > 0$. For any $\epsilon > 0$, it holds with probability greater than $1 - \epsilon$

$$\rho(G) \leq \rho^{ER} := (N - 1)p + \sqrt{2(N - 1)\, p\, (1 - p) \log \frac{2N}{\epsilon}} + \frac{2}{3} \log \frac{2N}{\epsilon}. \tag{5}$$

**Lemma 3.** Consider the symmetric adjacency matrix $G \in \mathbb{R}^{N \times N}$ of a network drawn from the SB model

$$\begin{cases} [G]_{kl} \sim \texttt{Bernoulli}(p_c), & \text{if } k \text{ and } l \in c, \\ [G]_{kl} \sim \texttt{Bernoulli}(q), & \text{if } k \in c \text{ and } l \in c' \neq c, \\ [G]_{kk} = 0, & \text{for } 1 \leq k \leq N, \end{cases}$$

where $\{[G]_{kl} : 1 \leq k < l \leq N\}$ are independent. For any $\epsilon > 0$, it holds with probability greater than $1 - \epsilon$

$$\rho(G) \leq \rho^{SB} =: (N - N/C)q + (N/C - 1)p_{\max} +$$
$$+ \sqrt{2\left((N - N/C)q(1 - q) + (N/C - 1)\sigma_{p,\max}^2\right) \log \frac{2N}{\epsilon}} + \frac{2}{3} \log \frac{2N}{\epsilon}.$$

where $p_{\max} = \max\{p_1, \ldots, p_C\}$ and $\sigma_{p,\max} = \max\{\sqrt{p_1(1 - p_1)}, \ldots, \sqrt{p_C(1 - p_C)}\}$.

Together with Lemma 6, the two results above allow us to establish a link between the edge connection probabilities in the graph—and thus the expected node degree—and the convergence of (QLD).

**Theorem 1.** Let $\epsilon > 0$ and $\mathcal{G}$ a network polymatrix game satisfying Assumption 1 with corresponding network drawn from the ER model with parameter $p$. If

$$T_k(N) > \rho^{ER} \text{ for all agents } k \in \mathcal{N},$$

then with probability at least $1 - \epsilon$, there is a unique QRE $\mathbf{x}^* \in \Delta$ and, for any initial condition, the (QLD) trajectories converge to $\mathbf{x}^*$.

**Theorem 2.** Let $\epsilon > 0$ and $\mathcal{G}$ a network polymatrix game satisfying Assumption 1 with corresponding network drawn from the SB model with probabilities $p_1, \ldots, p_C, q$. If

$$T_k(N) > \rho^{SB} \text{ for all agents } k \in \mathcal{N},$$

then with probability at least $1 - \epsilon$, there is a unique QRE $\mathbf{x}^* \in \Delta$ and, for any initial condition, the (QLD) trajectories converge to $\mathbf{x}^*$.

*Remark.* In both the ER and SB model, the the dominant $O(N)$ terms in the spectral-radius bounds are governed by expected node degrees. Indeed, the leading term in $\rho^{ER}$, namely $(N - 1)p$, equals the expected degree in the ER model. Likewise, the leading term in $\rho^{SB}$, namely

$$(N - N/C)q + (N/C - 1)p_{\max},$$

represents the maximum expected degree across all blocks in the SB network. Further, the sub-leading $O(\sqrt{N \log N})$ term is governed by the variances of the degree distributions: $p(1 - p)$ in the ER case, and on the block-wise maximum variance $\sigma_{p,\max}$ and variance $q(1 - q)$ in the SBM case.

**Implications to learning in multi-agent games** Theorems 1 and 2 show that (QLD) converges with low exploration rates whenever the expected degree of nodes in the network is small. Importantly, previous works (Sanders et al., 2018) linked the onset of unstable dynamics to the total number of agents $N$, but only in the limited case where all players interact with one-another, i.e., $p = 1$. Our bounds refine this finding by showing that the onset of instability is related to the expected degree in the network, not to $N$ explicitly. Further, once we control the expected degree, the variance terms are automatically controlled as well. As a consequence, the threshold for chaotic behavior increases from linear to exponential in $N$. In other words, with controlled expected degrees, chaotic dynamics can only emerge at exponentially large network sizes, far beyond the regimes considered in earlier analyses.

## 4 EXPERIMENTS

The aim of this section is to illustrate the implications of Theorems 1 and 2. We simulate the Q-Learning algorithm on network games, where the network is drawn from the ER and SB models. Note that we simulate the discrete-time algorithm described in Section 2, rather than integrating the continuous-time model (QLD). This is to explore the agreement between the theoretical predictions and the discrete-time update, which is the algorithm that is applied in practice. Each experiment runs for 4000 iterations, and we assess convergence numerically, as explained in Appendix E.1. We show that, as the expected node degree of the network increases, higher exploration rates are required for Q-Learning to convergence by finding the boundary between stable and unstable dynamics. We analyse how the boundary evolves with $N$ and provide empirical evidence that our results extend beyond the assumptions used to derive the bounds. In our experiments, we assume all agents have the same exploration rate $T$ and thus omit the dependency on $k$. As our results are lower bounds across all $k$, this is without loss of generality.

**Game Models.** We first simulate Q-Learning in the Network Shapley and Network Sato games, which are extensions of classical bimatrix games, introduced in Shapley (1964) and Sato et al. (2002) respectively, to the network polymatrix setting. In both cases, each edge defines the same bimatrix game, $(A^{kl}, A^{lk}) = (A, B) \ \forall (k, l) \in \mathcal{E}$ where in the Network Shapley game:

$$A = \begin{pmatrix} 1 & 0 & \beta \\ \beta & 1 & 0 \\ 0 & \beta & 1 \end{pmatrix}, B = \begin{pmatrix} -\beta & 1 & 0 \\ 0 & -\beta & 1 \\ 1 & 0 & -\beta \end{pmatrix},$$

for $\beta \in (0, 1)$; and in the Network Sato game:

$$A = \begin{pmatrix} \epsilon_X & -1 & 1 \\ 1 & \epsilon_X & -1 \\ -1 & 1 & \epsilon_X \end{pmatrix}, B = \begin{pmatrix} \epsilon_Y & -1 & 1 \\ 1 & \epsilon_Y & -1 \\ -1 & 1 & \epsilon_Y \end{pmatrix},$$

where $\epsilon_X, \epsilon_Y \in \mathbb{R}$. Fix $\epsilon_X = 0.5, \epsilon_Y = -0.3$ and $\beta = 0.2$. We also study the Conflict Network game, proposed in Ewerhart & Valkanova (2020). A network polymatrix game is a Conflict Network game if each bimatrix game $(A^{kl}, A^{lk})$ satisfies $(A^{kl})_{ij} = v^k (P^{kl})_{ij} - c_i^{kl}$ and $(A^{lk})_{ji} = v^j (P^{lk})_{ji} - c_j^{lk}$, where $v_k, v_l > 0$, $c_{kl} \in \mathbb{R}^{n_k}, c_{lk} \in \mathbb{R}^{n_l}$ and $(P^{kl})_{ij} + (P^{lk})_{ji} = 1$ for all $i \in S_k$ and $j \in S_l$. To generate Conflict Network games, we independently sample each $v_k$ from the uniform distribution over $[0, 1]$ and generate $P^{kl}$ by randomly sampling its elements from the uniform distribution over $[-5, 5]$. We then construct $(A^{kl}, A^{lk})$ such that the Conflict Network condition is satisfied.

**Erdős-Rényi Model.** In Figure 1, we examine the convergence properties of Q-Learning on networks generated from the Erdős-Rényi (ER) model. We vary the edge probability $p$, the exploration rate $T$, and the number of agents $N$. Figure 2 extends this analysis by plotting the convergence boundary as $N$ increases for different values of $p$. The key observation is that the rate of boundary growth increases with $p$, demonstrating that network density—parameterised by the edge probability $p$—has a direct impact on learning dynamics. Our results reveal that dense networks (high expected node degree) require higher exploration rates to guarantee convergence, consistent with previous findings in Sanders et al. (2018). Recall from Section 2, however, that higher exploration rates yield Quantal Response Equilibria (QRE) that deviate further from the Nash Equilibrium. Crucially, we find that this effect is substantially reduced in sparse networks (small $p$).

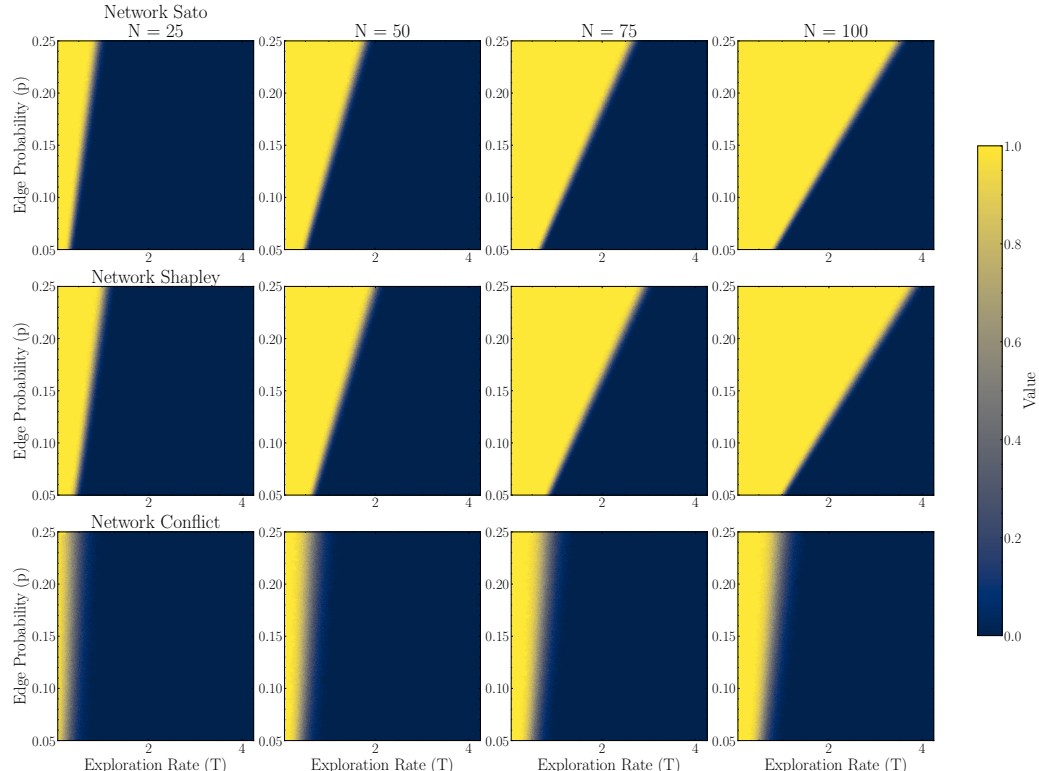

Figure 1: Proportion of (QLD) simulations that diverge in network games with network drawn from the Erdős-Rényi model using varying exploration rates $T$, edge probabilities $p$, and numbers of agents $N$. Each heatmap uses $(T, p)$ values on the grid $[0.05, 4.25] \times [0.05, 0.25]$. Convergence at low exploration rates is more likely in sparser networks, i.e., low $p$. Further, the boundary between convergent and divergent behaviour shifts rapidly with the number of agents $N$ when $p$ is high, highlighting the need to control $p$ for large $N$. Additional heatmaps are available in Appendix E.2.

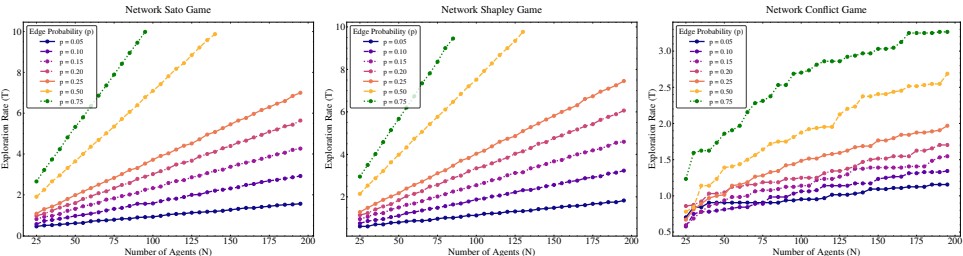

Figure 2: Variation of the empirical convergence boundary in various network games as $N$ increases, for different values of $p$. The $y$-axis shows the smallest exploration rate for which **all** 50 simulations converged. Dense networks (high $p$) require substantially higher exploration rates to ensure convergence, while sparse networks (low $p$) maintain low convergence thresholds even for large $N$. Recall that lower exploration rates are preferable so that the QRE is closer to the Nash equilibrium.

The Network Sato game in Figure 2 illustrates this phenomenon clearly. For $p = 0.05$, (QLD) converges for $T < 2.0$ even with over 150 agents. Conversely, for $p = 0.75$, convergence may fail with as few as 25 agents. These results also hold in the Conflict Network game, which extends beyond the setting of Assumption 1. Although there is some stochasticity in the boundary due to randomised payoffs, the fundamental trend remains: convergence occurs at lower exploration rates when the expected node degree is lower. These findings underscore the importance for practitioners of carefully designing network structures to ensure multi-agent learning produces stable, near-optimal equilibria.

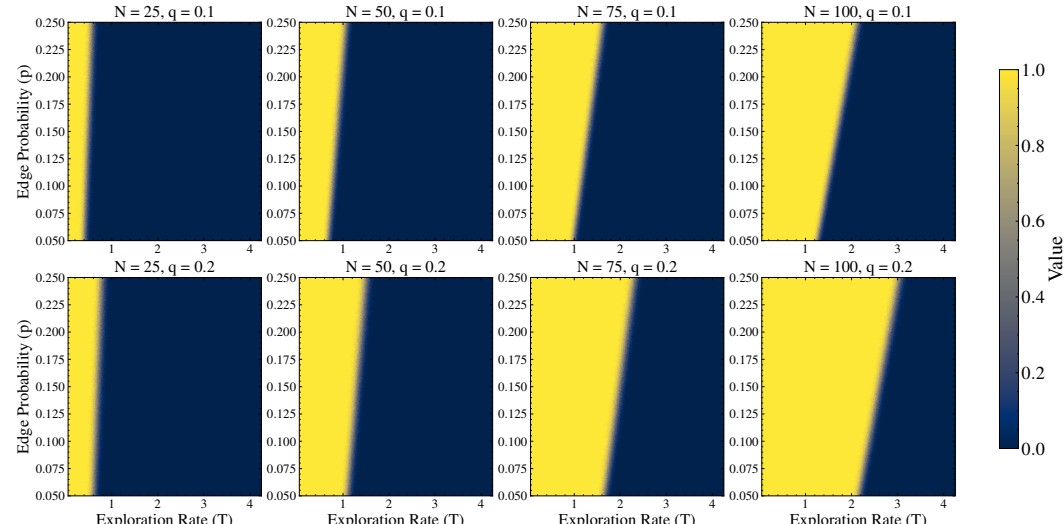

Figure 3: Proportion of (QLD) simulations that diverge in Network Sato games with network drawn from the Stochastic Block model, varying exploration rates $T$, intra-community edge probability $p$, inter-community edge probability $q$ and number of agents $N$. We use equidistant $(T, p)$ values in $[0.05, 4.25] \times [0.05, 0.25]$. The convergence boundary is jointly controlled by $p$ and $q$, offering greater flexibility to achieve convergence in structured networks. Extra results are provided in Appendix E.2.

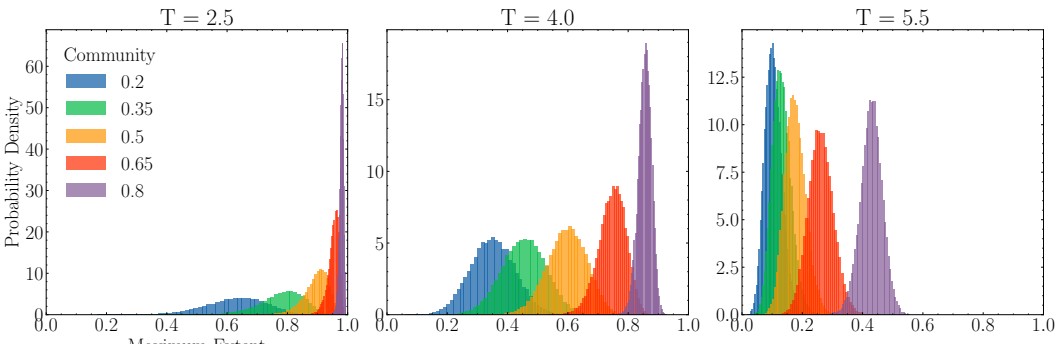

Figure 4: Probability density function of final strategy variation in Network Sato games on heterogeneous stochastic block networks with $N = 200$ agents, showing the maximum strategy variation across agents during the final 300 iterations, computed from 1024 independent simulations. Networks contain five communities with varying intra-community connection probabilities $p$ (shown in legend) and fixed inter-community probability $q = 0.1$. Communities with lower connectivity (blue, green) achieve convergence at lower exploration rates than densely connected communities (red, purple).

**Stochastic Block Model.** We next evaluate Q-Learning on networks generated from the Stochastic Block (SB) model. Figure 3 shows the proportion of diverged experiments in the Network Sato game, with results for the Network Shapley and Conflict games provided in Appendix E.2. Because the SB model has many parameters, we fix the number of agents in each community to $N_c = 5$ and suppose that $p_c = p$ for all communities $c$. We then assess the convergence of (QLD) for different values of $p$ and $T$, while varying $q$ and $N$. We find that, although each parameter influences last-iterate behaviour, convergence occurs less frequently in denser graphs, i.e., as $p$ and $q$ increase. Finally, we examine the case in which $p_c$ varies across communities in Figure 4. We simulate 1024 independent runs of QLD dynamics and compute the maximum difference between strategy components over the final iterations. The results show that communities with lower $p_c$ tend to exhibit smaller maximum differences. This suggests that allowing agents in different communities to use different exploration rates may yield convergence with smaller rates than required by Theorem 2.

## 5 CONCLUSION

This paper examined Q-Learning dynamics in network polymatrix games through the lens of random networks. In doing so, we showed an explicit relationship between the last-iterate convergence of Q-Learning and the expected degree of nodes in the underlying network. We provided theoretical lower bounds on the exploration rates required for convergence in network polymatrix games where the network is drawn from the Erdős-Rényi or Stochastic Block models and showed that the convergence may be guaranteed in games with many agents, so long as interactions in the network are controlled.

**Future Work.** In our experiments, we found that the implications of our theorems hold in settings beyond the assumptions under which they were derived. This suggests several directions for future work. One is to examine if our results generalise to broader classes of network games, e.g., games with continuous action sets. In addition, whilst our work considers the repeated play of a matrix game, extending these results to the *Markov Game* framework—which introduces a state variable—would enable theoretical guarantees to be placed in real-world multi-agent systems. Of further interest is the extension to other random network models, e.g., the Barabási-Albert model, used to model systems such as the Internet that exhibit preferential attachment. Notably, a direct extension to our current proof technique may be applied to random networks with dependent entries (see end of Appendix C). Analysing these models could reveal deeper connections between network structure and the convergence of learning algorithms.

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
