## APPENDIX

The appendix is organized as follows:

- Appendix A presents the necessary preliminaries for our proofs, including definitions and properties of *monotone games* and certain properties of matrix norms.

- Appendix B contains the proof of Lemma 1.

- Appendix C contains the proofs of Lemmas 2 and 3 and establishes asymptotic upper bounds on the spectral radius of networks drawn from the Erdős-Rényi and Stochastic Block models.

- Appendix D describes how payoffs are assigned to edges in the network polymatrix game under Assumption 1. We clarify that the underlying network is undirected.

- Appendix E contains further simulation studies and gives details on the computational schemes used to produce the visualizations from the main body.

- Appendix F gives details on the continuous-time Q-Learning Dynamic (QLD) and its relation to the underlying *independent Q-Learning* algorithm discussed in Section 2.

**Disclaimer:** LLMs were used in this work for grammar checks and generally polishing the text and code.

## A VARIATIONAL INEQUALITIES AND MONOTONE GAMES

The main idea of our convergence proof is to show that, under the conditions (4), the corresponding network game is a *strictly monotone* game Melo (2018); Parise & Ozdaglar (2019); Hadikhanloo et al. (2022); Sorin & Wan (2016). In such games, it is known that the equilibrium solution is unique Melo (2021); Facchinei & Pang (2004). It is further known, from Hussain et al. (2023), that (QLD) converges asymptotically in monotone games. We leverage this result to prove Lemma 1.

We begin by framing game theoretic concepts in the language of variational inequalities.

**Definition 4** (Variational Inequality). Consider a set $\mathcal{X} \subset \mathbb{R}^d$ and a map $F : \mathcal{X} \to \mathbb{R}^d$. The Variational Inequality (VI) problem $VI(\mathcal{X}, F)$ is given as

$$\langle \mathbf{x} - \mathbf{x}^*, F(\mathbf{x}^*) \rangle \geq 0, \qquad \text{for all } \mathbf{x} \in \mathcal{X}. \tag{6}$$

We say that $\mathbf{x}^* \in \mathcal{X}$ belongs to the set of solutions to a variational inequality problem $VI(\mathcal{X}, F)$ if it satisfies (6).

We now wish to reformulate the problem of finding Quantal Response Equilibria, or Nash Equilibria, as a problem of solving a variational inequality of a particular form. In such a case, the set $\mathcal{X}$ is identified with the joint simplex $\Delta$. The map $F$ is identified with the *pseudo-gradient* map of the game.

**Definition 5** (Pseudo-Gradient Map). The $\mathcal{G}$ be a network polymatrix game with payoff functions $(u_k)_{k \in \mathcal{N}}$. Then, the pseudo-gradient map of $\mathcal{G}$ is $F : \mathbf{x} \mapsto (-D_{\mathbf{x}_k} u_k(\mathbf{x}_k, \mathbf{x}_{-k}))_{k \in \mathcal{N}}$.

This reformulation has been used, for example, by Melo (2021) to show that a QRE of a game can be found by solving a variational inequality of a particular form. We reformulate their theorem for the particular case of network polymatrix games.

**Lemma 4** (Melo (2021)). Consider a game $\mathcal{G} = (\mathcal{N}, \mathcal{E}, (\mathcal{A}_k)_{k \in \mathcal{N}}, (A^{kl}, A^{lk})_{(k,l) \in \mathcal{E}})$ and for any $T_1, \ldots, T_N > 0$, let the *regularised* game $\mathcal{G}^H$ be the network game in which the payoff $u_k^H$ to each agent $k$ is given by

$$u_k^H(\mathbf{x}_k, \mathbf{x}_{-k}) = \sum_{l:(k,l) \in \mathcal{E}} \mathbf{x}_k^\top A^{kl} \mathbf{x}_l - T_k \langle \mathbf{x}_k, \ln \mathbf{x}_k \rangle. \tag{7}$$

Now, let $F^H$ be the pseudo-gradient map of $\mathcal{G}^H$. Then $\mathbf{x}^* \in \Delta$ is a QRE of $\mathcal{G}$ if and only if $\mathbf{x}^*$ is a solution to $VI(\Delta, F^H)$. .

In fact, this same lemma can be used to show that $\mathbf{x}^*$ is a QRE of $\mathcal{G}$ if and only if it is a Nash Equilibrium of $\mathcal{G}^H$. This concept of regularised games has also been used to show connections between the replicator dynamics Maynard Smith (1974) and Q-Learning dynamics in Leonardos & Piliouras (2022), and to design algorithms for Nash Equilibrium seeking Gemp et al. (2022). We require one final component from the study of variational inequalities which is often used to study uniqueness of equilibrium solutions: monotonicity Parise & Ozdaglar (2019); Melo (2018).

**Definition 6** (Monotone Games). Let $\mathcal{G}$ be a network polymatrix game with pseudo-gradient map $F$. $\mathcal{G}$ is said to be

1. *Monotone* if, for all $\mathbf{x}, \mathbf{y} \in \Delta$,
$$\langle F(\mathbf{x}) - F(\mathbf{y}), \mathbf{x} - \mathbf{y} \rangle \geq 0.$$

2. *Strictly Monotone* if, for all $\mathbf{x}, \mathbf{y} \in \Delta$,
$$\langle F(\mathbf{x}) - F(\mathbf{y}), \mathbf{x} - \mathbf{y} \rangle > 0.$$

3. *Strongly Monotone* with constant $\alpha > 0$ if, for all $\mathbf{x}, \mathbf{y} \in \Delta$,
$$\langle F(\mathbf{x}) - F(\mathbf{y}), \mathbf{x} - \mathbf{y} \rangle \geq \alpha \|\mathbf{x} - \mathbf{y}\|_2^2.$$

By proving monotonicity properties of a game, we can leverage the following results from literature.

**Lemma 5** (Melo (2021)). Consider a game $\mathcal{G}$ and for any $T_1, \ldots, T_N > 0$, let $F$ be the pseudo-gradient map of $\mathcal{G}^H$. $\mathcal{G}$ has a unique QRE $\mathbf{x}^* \in \Delta$ if $\mathcal{G}^H$ is strongly monotone with any $\alpha > 0$.

**Lemma 6** (Hussain et al. (2023)). If the game $G$ is *monotone*, then the Q-Learning Dynamics (QLD) converges to a unique QRE $\mathbf{x}^* \in \Delta$ with any positive exploration rates $T_1, \ldots, T_N > 0$.

Finally, note that a map $g : \Delta \to \mathbb{R}$ is *strongly convex* with constant $\alpha$ if, for all $\mathbf{x}, \mathbf{y} \in \Delta$

$$g(\mathbf{y}) \geq g(\mathbf{x}) + Dg(\mathbf{x})^\top (\mathbf{y} - \mathbf{x}) + \frac{\alpha}{2} \|\mathbf{x} - \mathbf{y}\|_2^2.$$

If the map $g(\mathbf{x})$ is twice-differentiable, then it is strongly convex if its Hessian $D_{\mathbf{x}}^2 g(\mathbf{x})$ is strongly positive definite with constant $\alpha$. Thus, all eigenvalues of $D_{\mathbf{x}}^2 g(\mathbf{x})$ are larger than $\alpha$. It also holds that, if $D_{\mathbf{x}}^2 g(\mathbf{x})$ is strongly positive definite, the gradient $D_{\mathbf{x}} g(\mathbf{x})$ is strongly monotone. To apply this in our setting, we use the following result.

**Proposition 1** (Melo (2021)). The function $g(\mathbf{x}_k) = T_k \langle \mathbf{x}_k, \ln \mathbf{x}_k \rangle$ is strongly convex with constant $T_k$.

### A.1 MATRIX NORMS

We close with a few properties of matrices that are useful towards our decomposition of the payoff matrices and the graph adjacency matrix.

**Proposition 2.** For any matrix $A$, $\|A\|_2 = \sqrt{\lambda_{\max}(A^\top A)}$, where $\lambda_{\max}(\cdot)$ denotes the largest eigenvalue of a matrix. If, in addition, the matrix is symmetric, $\|A\|_2 = \rho(A)$, where $\rho(A)$ is the spectral radius of $A$.

**Proposition 3** (Weyl's Inequality). Let $J = D + N$ where $D$ and $N$ are symmetric matrices. Then it holds that

$$\lambda_{\min}(J) \geq \lambda_{\min}(D) + \lambda_{\min}(N),$$

where $\lambda_{\min}(\cdot)$ denotes the smallest eigenvalue of a matrix.

**Proposition 4.** Let $G, A$ be matrices and let $\otimes$ denote the Kronecker product. Then

$$\|G \otimes A\|_2 = \|G\|_2 \|A\|_2 \tag{8}$$

**Proposition 5.** Let $A$ be a symmetric matrix. Then

$$|\lambda_{\min}(A)| \leq \rho(A)$$

**Lemma 7.** Let $A, B \in \mathcal{M}_{m,n}(\mathbb{R})$ such that $0 \leq A_{ij} \leq B_{ij}$ for any $1 \leq i \leq m$, $1 \leq j \leq n$. Then

$$\|A\|_2 \leq \|B\|_2.$$

*Proof.* Let $\mathbb{R}_+^n$ be the set of $n$-dimensional nonnegative vectors, i.e. with nonnegative entries. As $A$ and $B$ have nonnegative entries, we deduce that

$$\|A\|_2 = \sup_{\mathbf{x} \in \mathbb{R}^n} \|A\mathbf{x}\|_2 = \sup_{\mathbf{x} \in \mathbb{R}_+^n} \|A\mathbf{x}\|_2,$$

$$\|B\|_2 = \sup_{\mathbf{x} \in \mathbb{R}^n} \|B\mathbf{x}\|_2 = \sup_{\mathbf{x} \in \mathbb{R}_+^n} \|B\mathbf{x}\|_2.$$

Further, for any $\mathbf{x} \in \mathbb{R}_+^n$, it holds that

$$\|B\mathbf{x}\|_2 = \|(B - A)\mathbf{x} + A\mathbf{x}\|_2 \geq \|A\mathbf{x}\|_2,$$

as $(B - A)\mathbf{x}$ is nonegative. This increases the norm of any nonnegative vector $A\mathbf{x}$. Taking supremum in the above equation, we obtain that

$$\sup_{\mathbf{x} \in \mathbb{R}^n} \|A\mathbf{x}\|_2 = \sup_{\mathbf{x} \in \mathbb{R}_+^n} \|A\mathbf{x}\|_2 \leq \sup_{\mathbf{x} \in \mathbb{R}_+^n} \|B\mathbf{x}\|_2 = \sup_{\mathbf{x} \in \mathbb{R}^n} \|B\mathbf{x}\|_2.$$

$\square$

## B  PROOF OF LEMMA 1

The proof takes the following steps. We first decompose the derivative of the pseudo-gradient, which we dub the *pseudo-jacobian*, of the regularised game $\mathcal{G}^H$ into a term involving payoffs and a term involving exploration rates. In doing so, we can determine how the exploration rates should be balanced so that the pseudo-jacobian is positive definite, which yields monotonicity of the game. We then further decompose the payoff term into terms involving the payoff matrices and the network adjacency matrix. This exposes the connection between each of the three quantities: exploration rate, payoff matrices and network connectivity.

*Proof of Lemma 1.* Let $F$ be the pseudo-gradient of the regularised game $\mathcal{G}^H$. We define the pseudo-jacobian as the derivative of $F$, given by

$$[J(\mathbf{x})]_{k,l} = D_{\mathbf{x}_l} F_k(\mathbf{x})$$

It holds that if $\frac{J(\mathbf{x}) + J^\top(\mathbf{x})}{2}$ is positive definite for all $\mathbf{x} \in \Delta$ then $F(\mathbf{x})$ is monotone. We decompose $J$ as

$$J(\mathbf{x}) = D(\mathbf{x}) + N(\mathbf{x}),$$

where $D(\mathbf{x})$ is a block diagonal matrix with $-D^2_{\mathbf{x}_k \mathbf{x}_k} u_k^H(\mathbf{x}_k, \mathbf{x}_{-k})$ along the diagonal. $N(\mathbf{x})$ is an off-diagonal block matrix with

$$[N(\mathbf{x})]_{k,l} = \begin{cases} -D_{\mathbf{x}_k, \mathbf{x}_l} u_k^H(\mathbf{x}_k, \mathbf{x}_{-k}) & \text{if } (k,l) \in \mathcal{E} \\ \mathbf{0} & \text{otherwise} \end{cases}.$$

Now notice that $-u_k^H(\mathbf{x}_k, \mathbf{x}_{-k}) = T_k \langle \mathbf{x}_k, \ln \mathbf{x}_{-k} \rangle - \sum_{l:(k,l) \in \mathcal{E}} \mathbf{x}_k^\top A^{kl} \mathbf{x}_l$. Therefore, $D(x)$ is simply the Hessian of the entropy regularisation term $T_k \langle \mathbf{x}_k, \ln \mathbf{x}_k \rangle$. From Proposition 1, it holds then that $D(\mathbf{x})$ is strongly positive definite with constant $T = \min_k T_k$. Let $\bar{J}(\mathbf{x})$ be defined as

$$\bar{J}(\mathbf{x}) = D(\mathbf{x}) + \frac{N(\mathbf{x}) + N^\top(\mathbf{x})}{2}.$$

In words, $\bar{J}(\mathbf{x})$ is the symmetric component of $J(\mathbf{x})$. We may now use Weyl's inequality to write

$$\lambda_{\min}(\bar{J}(\mathbf{x})) \geq \lambda_{\min}(D(\mathbf{x})) + \lambda_{\min}\left(\frac{N(\mathbf{x}) + N(\mathbf{x})^\top}{2}\right)$$

$$\geq T - \rho\left(\frac{N(\mathbf{x}) + N(\mathbf{x})^\top}{2}\right)$$

$$= T - \frac{1}{2}\|N(\mathbf{x}) + N(\mathbf{x})^\top\|_2$$

To determine $\|N(\mathbf{x}) + N(\mathbf{x})^\top\|_2$, we first notice that, in network polymatrix games each block of $N(\mathbf{x})$ is given by

$$[N(\mathbf{x})]_{k,l} = \begin{cases} -A^{kl} & \text{if } (k,l) \in \mathcal{E} \\ \mathbf{0} & \text{otherwise.} \end{cases}$$

whilst

$$[N(\mathbf{x})^\top]_{k,l} = \begin{cases} -(A^{lk})^\top & \text{if } (k,l) \in \mathcal{E} \\ \mathbf{0} & \text{otherwise.} \end{cases}$$

To write this in the form of a kronecker product, we leverage Assumption 1, namely that each edge corresponds to the same bimatrix game with payoff matrices $(A, B)$. We decompose each edge into a half-edge along which $A$ is played, and a half-edge along which $B$ is played. In doing so, we may decompose the adjacency matrix $G = G_{k \to l} + G_{l \to k}$. The non-zero elements of $G_{k \to l}$ correspond to half edges along which $A$ is played, $G_{l \to k}$ denote the half-edges along which $B$ is played. With this definition in place, we may write

$$N(\mathbf{x}) + N(\mathbf{x})^\top = -(A + B^\top) \otimes G_{k \to l} - (A^\top + B) \otimes G_{l \to k}$$

Then, by Proposition 4

$$\frac{1}{2}\|N(\mathbf{x}) + N(\mathbf{x})^\top\|_2 = \frac{1}{2}\|(A + B^\top) \otimes G_{k \to l} + (A^\top + B) \otimes G_{l \to k}\|_2$$

$$\leq \frac{1}{2}\|(A + B^\top) \otimes G_{k \to l}\|_2 + \frac{1}{2}\|(A^\top + B) \otimes G_{l \to k}\|_2$$

$$= \frac{1}{2}\|A + B^\top\|_2(\|G_{k \to l}\|_2 + \|G_{l \to k}\|_2)$$

$$\leq \|A + B^\top\|_2\|G\|_2$$

$$= \delta_I \rho(G)$$

where in the final inequality, we use Lemma 7. From this we may establish that

$$\lambda_{\min}(\bar{J}(x)) \geq T - \delta_I \rho(G)$$

Therefore, if $T > \delta_I \rho_G$, the pseudo-jacobian is strongly positive definite with constant $T - \delta_I \rho(G)$, from which it is established that $\mathcal{G}^H$ is strongly monotone with the same constant. From Lemma 5 Melo (2021) the QRE is unique, and from Lemma 6, it is asymptotically stable under (QLD). $\square$

## C    PROOFS OF LEMMAS 2 AND 3

We now focus on establishing upper bounds on the spectral radius when the network is drawn from the Erdős-Rényi or Stochastic Block models. The proof idea is to decompose $G$ into $\mathbb{E}[G]$ and $\tilde{G} = G - \mathbb{E}[G]$. Then, $\rho(\mathbb{E}[G])$ is deterministic and can be computed in closed form. For $\rho(\tilde{G})$, we require Lemma 9, which relies on Bernstein's matrix inequality (Tropp, 2015).

We structure this appendix as follows: first list the required results (Lemmas 8 and 9 and Corollary 1), then give proofs for Lemmas 2 and 3 and then the proof for the key Lemma 9. Finally, we comment on the possibility to extend these results to the dependent entry case.

**Lemma 8** (Bernstein). Let $Y^{(1)}, \ldots, Y^{(K)}$ be independent and symmetric $N \times N$ random matrices with zero-mean entries, i.e., $\mathbb{E}[Y_{ij}^{(k)}] = 0$ almost surely for any $1 \le i, j \le N$ and $1 \le k \le K$. Define

$$\tilde{G} = Y^{(1)} + \cdots + Y^{(K)},$$
$$v^2 = \rho\left(\mathbb{E}[\tilde{G}^2]\right) = \rho\left(\mathbb{E}[(Y^{(1)})^2 + \cdots + (Y^{(K)})^2]\right).$$

If $\rho(Y^{(k)}) \le L$ for all $1 \le k \le K$, then the following holds for any $t > 0$

$$P(\rho(\tilde{G}) > t) \le 2N \exp\left(\frac{-t^2/2}{v^2 + Lt/3}\right). \tag{9}$$

**Corollary 1.** We will use a different form of the above inequality, which follows by setting the RHS of equation 9 to $\epsilon$ and solving for $t$ in terms of $\epsilon$. For any $\epsilon > 0$, it holds that

$$\rho(\tilde{G}) \le \sqrt{2v^2 \log \frac{2N}{\epsilon}} + \frac{2L}{3} \log \frac{2N}{\epsilon} \text{ with probability at least } 1 - \epsilon.$$

**Lemma 9.** Let $G$ be the adjacency matrix of a random graph with 0s on the main diagonal and with independent entries $[G]_{ij} \sim \texttt{Bernoulli}(p_{ij})$, apart from $[G]_{ij}$ and $[G]_{ji}$, which are equal. Let $\tilde{G} = G - E[G]$ with entries $[\tilde{G}]_{ij} = [G]_{ij} - p_{ij}$. Then $\tilde{G}$ satisfies Corollary 1 with

$$v^2 = \max_{1 \le i \le N} \sum_{j \ne i} p_{ij}(1 - p_{ij})$$
$$L = \max_{1 \le i < j \le N} \{p_{ij}, 1 - p_{ij}\} \le 1.$$

For simplicity, as $L$ only appears in the non-dominant term, we use $L = 1$. In the ER case, this amounts to

$$\rho(\tilde{G}) \le \sqrt{2(N-1)\,p\,(1-p) \log \frac{2N}{\epsilon}} + \frac{2}{3} \log \frac{2N}{\epsilon} \text{ with probability at least } 1 - \epsilon.$$

In the SBM case, this amounts to

$$\rho(\tilde{G}) \le \sqrt{2\left((N - N/C)q(1-q) + (N/C - 1)\sigma_{\text{p,max}}^2\right) \log \frac{2N}{\epsilon}} + \frac{2}{3} \log \frac{2N}{\epsilon}$$

with probability at least $1 - \epsilon$ and where $\sigma_{\text{p,max}} = \max_{1 \le l \le C} \sqrt{p_l(1 - p_l)}$.

Proofs for Corollary 1 and Lemma 9 can be found at the end of the section.

*Proof of Lemma 2.* For the deterministic part, we have that

$$\mathbb{E}[G] = \begin{pmatrix} 0 & p & p & \cdots & p \\ p & 0 & p & \cdots & p \\ p & p & 0 & \cdots & p \\ \vdots & \vdots & \vdots & \ddots & \vdots \\ p & p & p & \cdots & 0 \end{pmatrix} = pJ_N - pI_N, \tag{10}$$

with eigenvalues $\lambda_1 = (N-1)p$ and $\lambda_2 = \ldots = \lambda_N = -p$. Hence $\rho(\mathbb{E}[G]) = (N-1)p$. Further, by applying Lemma 9, we obtain that with probability at least $1 - \epsilon$

$$\rho(G) \le \rho(\mathbb{E}[G]) + \rho\left(\tilde{G}\right) \le (N-1)p + \sqrt{2(N-1)\,p\,(1-p)\log\frac{2N}{\epsilon}} + \frac{2}{3}\log\frac{2N}{\epsilon}.$$

$\square$

*Proof of Lemma 3.* Let $I_N$ be the identity matrix and $J_N$ be the matrix whose elements are all ones. For the deterministic part, we have that

$$\mathbb{E}[G] = \begin{pmatrix} P_1 & Q & Q & \cdots & Q \\ Q & P_2 & Q & \cdots & Q \\ Q & Q & P_3 & \cdots & Q \\ \vdots & \vdots & \vdots & \ddots & \vdots \\ Q & Q & Q & \cdots & P_C \end{pmatrix},$$

where $Q = qJ_N$ and $P_l = pJ_m - pI_m$ for $1 \le l \le C$, as in equation 10. Thus

$$\rho\left(\mathbb{E}[G]\right) \le \|\mathbb{E}[G]\|_1 = (N - N/C)q + \max_{1 \le l \le C}\{(N/C - 1)p_l\} = (N - N/C)q + (N/C - 1)p_{\max}$$

$$= Nq + N(p_{\max} - q)/C - p_{\max},$$

where $p_{\max} = \max_{1 \le l \le C} p_l$. Let $\sigma_{\mathrm{p,max}} = \max\left\{\sqrt{p_1(1-p_1)}, \ldots, \sqrt{p_C(1-p_C)}\right\}$. By applying Lemma 9 for $\tilde{G}$ and combining the two upper bounds, we obtain that

$$\rho(G) \le \rho(\mathbb{E}[G]) + \rho\left(\tilde{G}\right) \le (N - N/C)q + (N/C - 1)p_{\max} +$$

$$\sqrt{2\left((N - N/C)q(1-q) + (N/C - 1)\sigma_{\mathrm{p,max}}^2\right)\log\frac{2N}{\epsilon}} + \frac{2}{3}\log\frac{2N}{\epsilon}.$$

$\square$

*Proof of Corollary 1.* Let $\epsilon = 2N\exp\left(\frac{-t^2/2}{v^2 + Lt/3}\right)$. Rearranging, we get a unique solution

$$t = \frac{L}{3}\log\frac{2N}{\epsilon} + \sqrt{\frac{L^2}{9}\left(\log\frac{2N}{\epsilon}\right)^2 + 2v^2\log\frac{2N}{\epsilon}}$$

We thus have that for any $\epsilon > 0$

$$P\left(\rho(A) > \frac{L}{3}\log\frac{2N}{\epsilon} + \sqrt{\frac{L^2}{9}\left(\log\frac{2N}{\epsilon}\right)^2 + 2v^2\log\frac{2N}{\epsilon}}\right) \le 1 - \epsilon.$$

For simplicity, we use the slightly looser bound

$$P\left(\rho(A) > \frac{2L}{3}\log\frac{2N}{\epsilon} + \sqrt{2v^2\log\frac{2N}{\epsilon}}\right) \le 1 - \epsilon,$$

obtained by applying $\sqrt{a^2 + b} < a + \sqrt{b}$ for $a, b \ge 0$, to the previous inequality. $\square$

*Proof of Lemma 9.* Notation-wise, let $e_i$ be the canonical basis vectors for $1 \le i \le N$. For each $1 \le i < j \le N$, define $Y^{ij} = [\tilde{G}]_{ij}(e_i e_j^T + e_j e_i^T)$, which keeps the $ij^{th}$ and $ji^{th}$ elements of $\tilde{G}$ and sets everything else to 0. Then

$$\tilde{G} = \sum_{1 \le i < j \le N} Y^{ij}.$$

By definition, the matrices $Y^{ij}$ are independent, symmetric random matrices and $\mathbb{E}[Y^{ij}] = 0$ for $1 \le i < j \le N$. We now determine $L$ and $v^2$.

For $L$, note that $Y_{ij}$ has two non-zero eigenvalues, i.e., $[\tilde{G}]_{ij} = [G]_{ij} - p_{ij}$ and $-[\tilde{G}]_{ij} = -[G]_{ij} + p_{ij}$ with corresponding eigenvectors $e_i + e_j$ and $e_i - e_j$, respectively. Thus, $\rho(Y^{ij})$ can only take the values $p_{ij}$ or $1 - p_{ij}$ and $L$ can be chosen as $\max_{1 \leq i < j \leq N}\{p_{ij}, 1 - p_{ij}\}$, or simpler, 1.

For $v^2$, note that $(Y^{ij})^2 = [\tilde{G}]_{ij}^2(e_ie_i^T + e_je_j^T) = ([G]_{ij} - p_{ij})^2(e_ie_i^T + e_je_j^T)$ , which is a diagonal matrix, and further $\mathbb{E}[(Y^{ij})^2] = p_{ij}(1 - p_{ij})(e_ie_i^T + e_je_j^T)$. Thus the matrix

$$\sum_{1 \leq i < j \leq N} \mathbb{E}[(Y^{ij})^2]$$

is diagonal, with the ii$^{th}$ entry given by $\sum_{j \neq i} p_{ij}(1 - p_{ij})$, hence $v^2 = \max_{1 \leq i \leq N} \sum_{j \neq i} p_{ij}(1 - p_{ij})$. Alternatively, one can compute $\mathbb{E}[\tilde{G}^2]$ directly and notice that the cross-diagonal entries are exactly 0 by the independence of the $Y^{ij}$ matrices for $1 \leq i < j \leq N$.

We now specialize to the ER and SBM cases. In the ER case, $\max_{1 \leq i \leq N} \sum_{j \neq i} p_{ij}(1 - p_{ij})$ is independent of $i$ and equals $(N - 1)p$. In the SBM case, we obtain a summation similar to the one from the proof of Lemma 3. Specifically, on the $i^{th}$ row we have $N/C - 1$ blocks of size $C$ where the edge connection is $q$, and one block of size $N/C$ where the edge connection is $p_i$. Thus

$$v^2 = \max_{1 \leq l \leq C}(N - N/C)q(1 - q) + (N/C - 1)p_l(1 - p_l).$$

$\square$

**Dependent entry case.** So far, we focused on random network models with independent entries, i.e., the ER and SB models. Extensions to the case of dependent entries are possible as long as we can impose constraints on the covariance between entries, by adopting a modified version a Bernstein's inequality, e.g., see Kallabis & Neumann (2006).

# D  ASSIGNMENT OF PAYOFFS TO EDGES

In this section, we clarify the assignment of payoff matrices $(A, B)$ to each edge. In particular, we begin with an *undirected graph* $(\mathcal{N}, \mathcal{E})$, in particular, one with a *symmetric* adjacency matrix $G$. For each edge $(k, l) \in \mathcal{E}$, we randomly assign either $A$ to $k$ and $B$ to $l$ or vice versa.

As an example, consider a 3 player network game where the adjacency matrix is given by

$$G = \begin{pmatrix} 0 & 1 & 0 \\ 1 & 0 & 1 \\ 0 & 1 & 0 \end{pmatrix}$$

From Assumption 1, it must be that both edges $(1, 2), (2, 3)$ are assigned the same payoff matrices $(A, B)$. One example is to assign the edge $(1, 2)$ to the payoffs $(B, A)$ and $(2, 3)$ to $(B, A)$. This yields the following payoffs for each agent

$$u_1 = x_1^\top B x_2$$

$$u_2 = x_2^\top A x_1 + x_2^\top B x_3$$

$$u_3 = x_3^\top A x_2.$$

Another choice is to assign the edge $(1, 2)$ to the payoffs $(A, B)$ and $(2, 3)$ to $(A, B)$.

$$u_1 = x_1^\top A x_2$$

$$u_2 = x_2^\top B x_1 + x_2^\top A x_3$$

$$u_3 = x_3^\top B x_2.$$

Finally, there is also the option to assign to the edge $(1, 2)$ the payoffs $(A, B)$ and $(2, 3)$ to $(B, A)$. This gives the payoffs

$$u_1 = x_1^\top A x_2$$

$$u_2 = x_2^\top B x_1 + x_2^\top B x_3$$

$$u_3 = x_3^\top A x_2.$$

Notice that, in any case, the underlying graph remains undirected, whilst the payoffs are randomly assigned and fixed throughout the game. This process can be conceptualised as randomly assigning a directed payoff matrix to each half-edge. Specifically, for each undirected edge $(k, l)$, we assign matrix $A$ to the directed half-edge $k \to l$ and matrix $B$ to the directed half-edge $l \to k$. Theorems 1 and 2 hold regardless of the ordering of the half-edges.

# E    FURTHER SIMULATION RESULTS

## E.1    SIMULATION SETUP

**Q-Learning hyperparameters** In all simulations, we iterate the Q-Learning algorithm defined in Equations 1 and 2. We use learning rate $\alpha_k = 0.1$ for all agents $k$. As shown in Tuyls et al. (2006), the $\alpha_k$ parameter amounts to a time rescaling in the continuous-time dynamic (QLD), so long as all agents use the same (constant) learning rate.

**Numerical convergence** In all simulations (e.g., Figure 1), we must evaluate numerically whether a Q-learning trajectory has converged or not. To this end, we run Q-learning for $4000$ steps and analyse the last $300$ steps. For these steps $300$, to which we will refer as the *trajectory*, we compute (i) the variance of each component of the trajectory and take the mean across components, and (ii) the componentwise relative difference, defined as the maximum across components of

$$\frac{\max(\text{trajectory}) - \min(\text{trajectory})}{\max(\text{trajectory})}$$

We consider a trajectory to have converged if the mean variance is below $10^{-2}$ and relative difference below $10^{-5}$. In Figures 4 and 8, we plot on the x-axis the maximum component of the absolute difference, defined for each agent $k$ as

$$\max_{i \in \mathcal{A}_k} \left\{ \max(\text{trajectory}_{ki}) - \min(\text{trajectory}_{ki}) \right\},$$

where $\text{trajectory}_{ki}$ refers to the mixed strategy of action $i$ for agent $k$.

**Computational resources** All experiments were run on an AMD Rome CPU cluster node with 128 cores and 2 GHz clock.

## E.2    ADDITIONAL SIMULATIONS

Figure 5 expands on Figure 1 by analysing network games with networks generated from the Erdős-Rényi model, with the number of agents $N$ varying from 15 to 50 and the edge probability $p$ over a wider range $(0.05, 1)$. By examining the boundary between convergent (blue) and divergent (yellow) regions, we find that convergent behaviours persist for low exploration rates only if $p$ is small. By contrast, for large $p$ (dense networks), the boundary shifts rapidly as $N$ increases.

Figure 7 extends Figure 3 by illustrating the convergence of (QLD) in Network Sato, Shapley, and Conflict games. Additionally, Figure 6 explores the impact of varying the number of agents from $N = 15$ to $N = 50$ and using the full range of $p \in (0.05, 1)$. This analysis shows that both $p$ and $q$ influence the location of the boundary separating convergent and non-convergent behaviours. This finding is notable because it also applies to the Network Conflict game, which lies outside the scope of Assumption 1

Finally, Figure 8 expands on Figure 4 by simulating Q-Learning on a Network Sato game with 250 agents and 1024 initial conditions. The results show that communities with low probability $p_c$ of intra-community edges exhibit less variation in the final 300 iterations compared to those with high $p_c$. This suggests that convergence is possible with lower exploration rates $T_k$ provided that heterogeneous exploration rates are employed, i.e., allowing for $T_k$ to be a function of $p_c$.

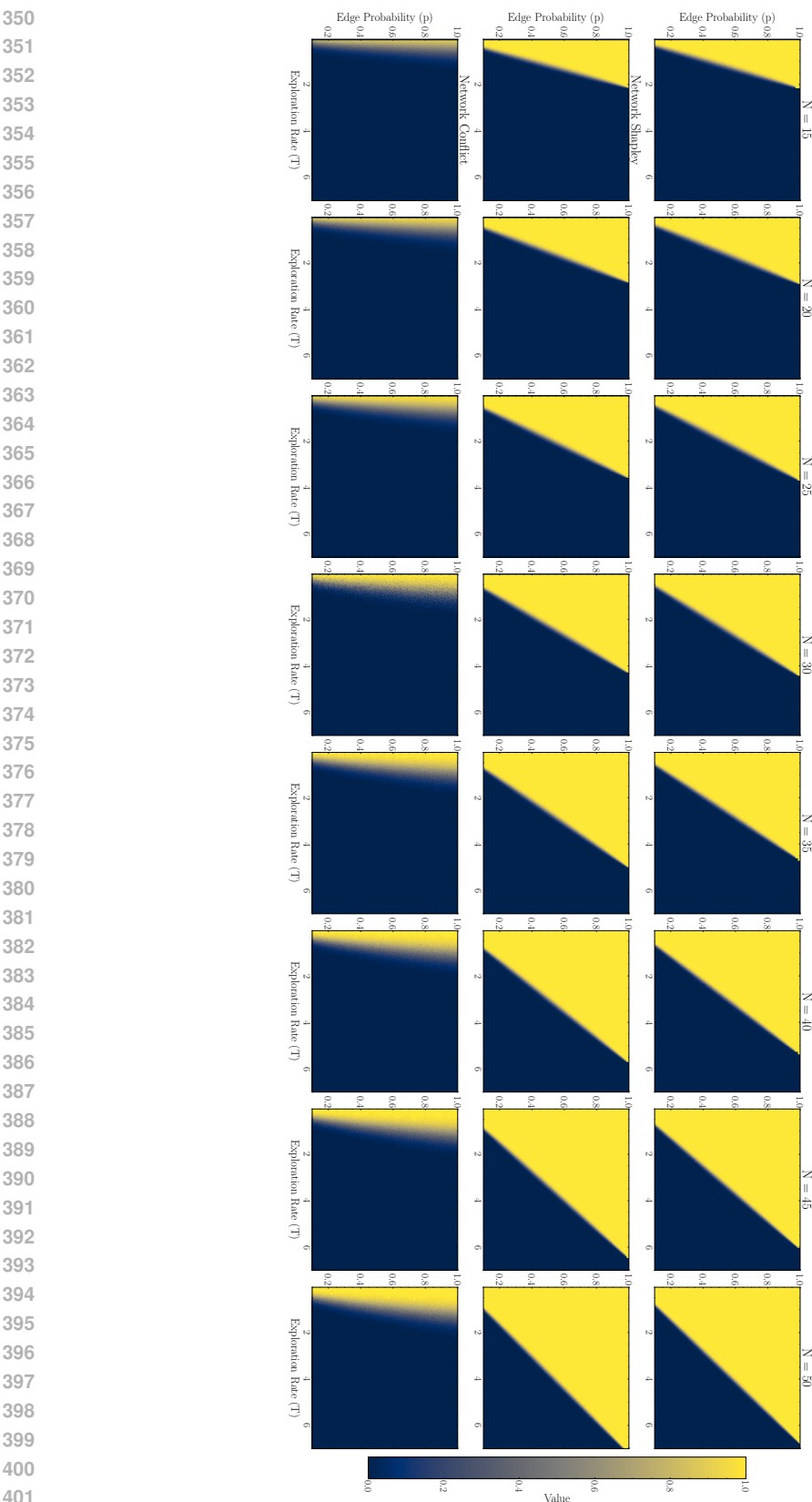

Figure 5: Proportion of (QLD) simulations that diverge in Network Sato, Shapley and Conflict games with networks drawn from the Erdős-Rényi model, varying exploration rates $T \in (0.05, 7)$, edge probability $p \in (0.05, 1)$, and number of agents $N \in \{15, 20, \ldots, 45, 50\}$. Because we are using the full range of $p$ values up to 1, we only display results up to $N = 50$.

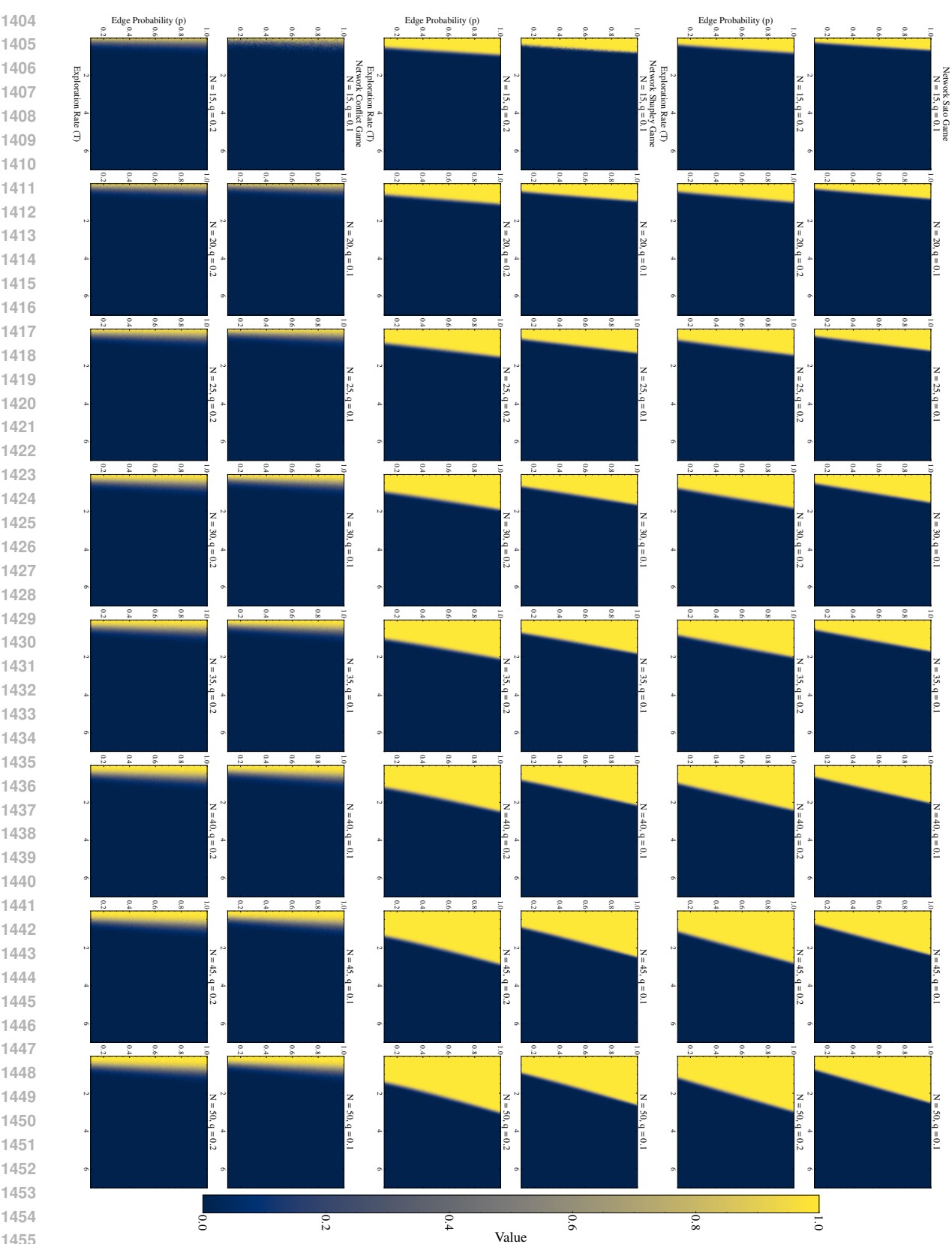

Figure 6: Proportion of (QLD) simulations that diverge in Network Sato, Shapley and Conflict games with networks drawn from the Stochastic Block model, varying exploration rates $T \in (0.05, 7)$, intra-community edge probability $p \in (0.05, 1)$, inter-community edge probability $q \in \{0.1, 0.2\}$ and number of agents $N \in \{15, 20, \ldots, 45, 50\}$. Because we are using the full range of $p$ values up to 1, we only display results up to $N = 50$.

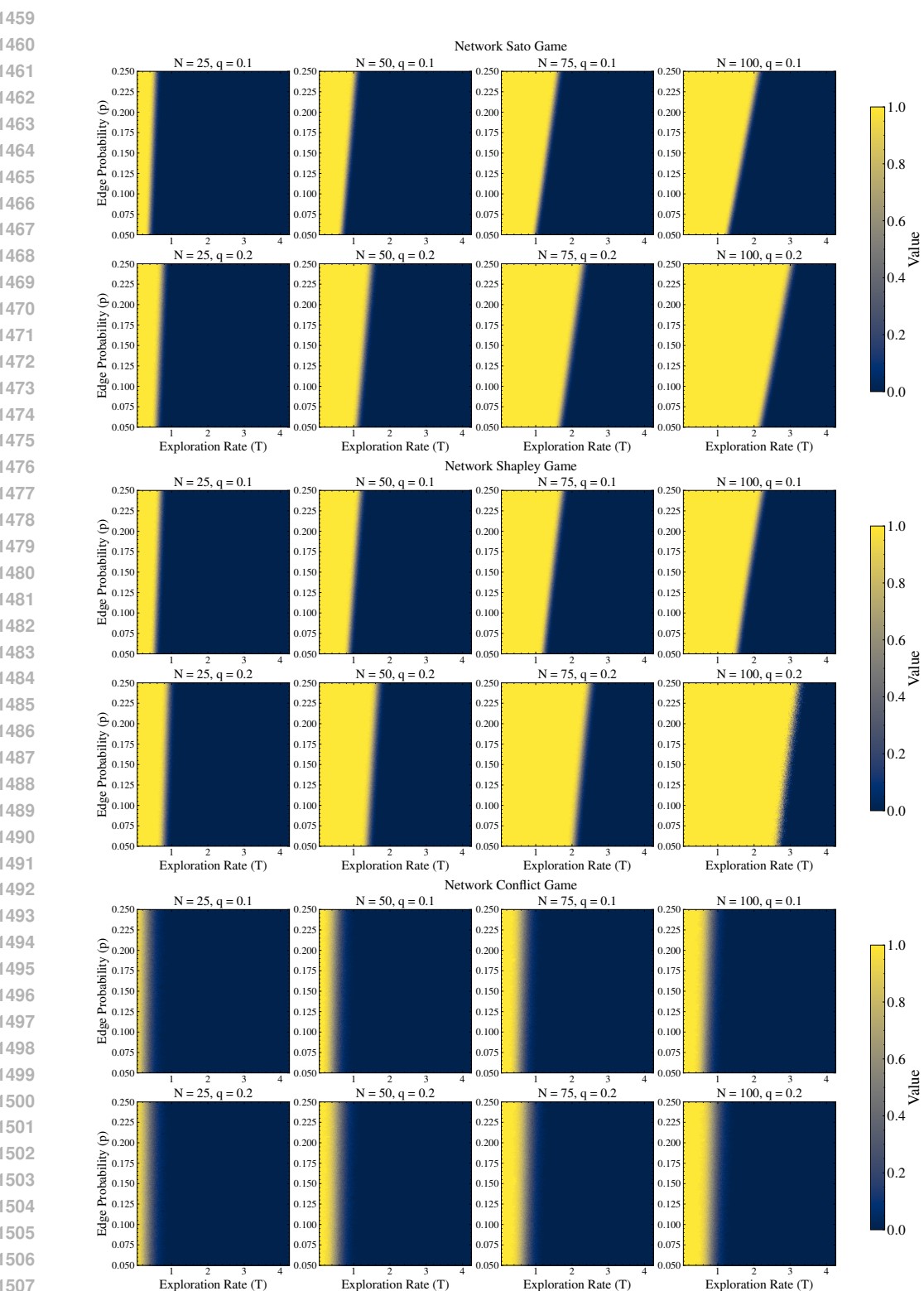

Figure 7: Continuation of Figure 6 for larger number of agents $N$ and a restricted $p$ range of $(0.05, 0.25)$, as used in the main body.

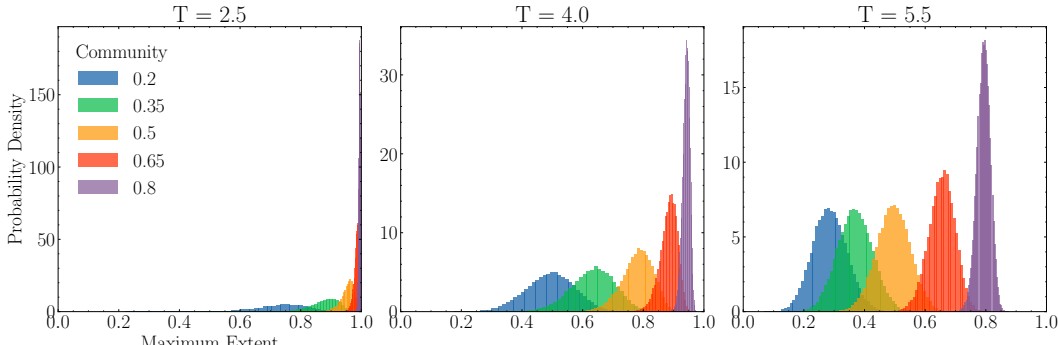

Figure 8: Probability density of final strategy variation in Network Sato games on heterogeneous stochastic block networks with $N = 250$ agents, showing the maximum strategy variation across agents during the final 300 iterations, computed from 1024 independent simulations. Networks contain five communities with varying intra-community connection probabilities $p$ (shown in legend) and fixed inter-community probability $q = 0.1$.

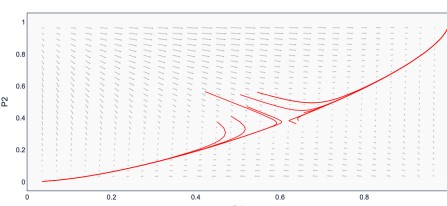 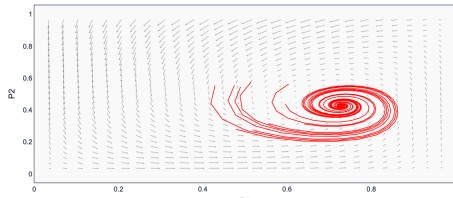

Figure 9: Trajectories of the Q-Learning Algorithm (red) plotted on top of the phase portrait (black) generated by (QLD) in two-player - two action games. The x-axis (resp. y-axis) denotes the probability with which the first (resp. second) player chooses their first action. In both cases, $T = 0.3$. The payoff matrices for the left and right image are given in Figures 12 and 14 of Tuyls et al. (2006) respectively.

## F  Q-LEARNING DYNAMICS

Our work is centered on independent (decentralised) multi-agent learning in normal-form games. In this setting we study the *independent q-learning* algorithm (Leslie & Collins, 2006). Here, each agent independently maintains a Q-value of each action $i \in \mathcal{A}_k$. Upon playing action $i$ at time step $t$ and receiving its associated reward $r_{ki}$, the agent updates its Q-value to be a weighted sum of its current estimation and the received reward. This is summarised in the update rule

$$Q_{ki}(t+1) = (1 - \alpha_k)Q_{ki}(t) + \alpha_k R_{ki}(t), \tag{11}$$

where $\alpha_k \in [0, 1]$ is a step size and $R_{ki}(t)$ refers to the random realisation of the payoff at time $t$. Notice that this realisation depends on the actions of other agents in the environment, although these actions are unknown to agent $k$.

Next, the agent updates their strategy according to their exploration policy using the *Boltzmann exploration* scheme, so that

$$x_{ki}(t) = \frac{\exp(Q_{ki}(t)/T_k)}{\sum_{j \in \mathcal{A}k} \exp(Q_{kj}(t)/T_k)} \tag{12}$$

By modifying the exploration rate $T_k$, the agent can smoothly move from high exploration, where the Q-values have little influence, to high exploitation. In this manner, Q-Learning presents a strong model whereby the influence of exploration can be examined.

In Sato & Crutchfield (2003) and Kaisers & Tuyls (2011), the authors apply tools from evolutionary game theory (EGT) (Hofbauer & Sigmund, 2003) to study multi-agent learning algorithms. To describe this approach, we first make a slight abuse of notation in which time $t$ will be regarded as a continuous time variable rather than discrete. Then, we can consider a learning algorithm to be a map from the joint strategy $\mathbf{x}(t)$ at time $t$ to $\mathbf{x}(t + \delta t)$ where $\delta t << 1$ is the time between consecutive updates. We will also take a deterministic approximation of the Q-update, by expressing it through the *expected* reward to agent $k$, given the opponents' joint state $\mathbf{x}_{-k}(t)$. This yields the update

$$Q_{ki}(t+1) = (1 - \alpha_k)Q_{ki}(t) + \alpha_k r_{ki}(\mathbf{x}_{-k}(t)). \tag{13}$$

If we take the limit as $\delta t \to 0$, we arrive at a continuous-time dynamical system that approximates the expected behaviour of the algorithm. As an example, we depict in Figure 9 traces of the Q-Learning update overlaid on its continuous time approximation, which we introduce subsequently. The advantage of this approach is that the tools of continuous dynamical systems can be used to prove properties of the idealisation, which in turn tells us about the algorithm itself. For the full derivation of this dynamic from the Q-Learning update see Appendix A.1 of Leonardos et al. (2024) or Section 3.2 of Tuyls et al. (2006).