# OpenReview forum: "Convergence and Connectivity: Asymptotic Dynamics of Multi-Agent Q-Learning in Random Networks"
_ICLR.cc/2026/Conference — Submitted to ICLR 2026_

### Official Review · Reviewer_ckUe · 2025-10-24

**Soundness:** 3
**Presentation:** 3
**Contribution:** 2
**Rating:** 4
**Confidence:** 4

**Summary:**

The paper studies the convergence of Q-learning dynamics (QLD) - a model of continuous time dynamics which balances exploitation with exploration through a parameter T_k for each player k - in random networks, i.e., networks in which nodes are agents (players) and games between agents are played if an edge is randomly assigned between them. For the theoretical results, the paper assumes that the game across all edges is the same, i.e., a bimatrix game with matrices A, B. The paper shows that a sufficient condition for QLD to converge is that the exploration parameter is large enough for all players in the network, and specifically if it is larger than the product of the highest eigenvalue of the adjacency matrix and a "similarity of payoffs" index. The paper sharpens this result in two settings of random networks - one in which each edge is assigned randomly with probability p and one in which nodes form clusters with probabilities of edges being different within and between these clusters. Finally, the paper contains experiments in 3 types of games which demonstrate that the sufficiency condition provided in the main results can be likely further generalised.

**Strengths:**

The paper is mathematically rigorous and the results are correct to the extent that I could verify. There are enough details to reproduce the experiments and the main result (Lemma 1 and its two instantiations in Theorems 1 and 2) are neat. The main finding that convergence is easier with less connections in the networks is also were formulated.

**Weaknesses:**

- The theoretical results are established under limiting assumptions, i.e., that the game played across each edge is the same. Experiments in a game with different payoffs across edges aim to demonstrate the contrary. However, the experimental and theoretical analyses as presented seem to not disentangle the effect of cooperation or competition in the underlying game from the other effects that are studied, i.e., exploration parameter, number of agents, and density of connections.
- The result that convergence of Q-learning ensues for higher exploration rates is already well-known in simpler settings and the technical contribution to establish this result in random networks doesn't seem to substantially generalise existing techniques or insights.

**Questions:**

- Can the authors provide a more detailed comparison to the predecessor paper of this work, Hussain et. al (2024)? In particular, I think
- For the Sato game, the similarity index should be high (or not?) since the payoffs are aligned, but convergence occurs even for low exploration rates. Does this imply that the sufficient condition in Lemma 1 can be significantly tightened in some cases?
- Related to the above, the authors could discuss more the choice of games and how they cover interesting cases.
- The paper mentions in the introduction (lines 79-80) about an apparent contrast with the finding in Sanders. That statement confused me and it would be great if they could clarify what they mean.

In general, I find the paper clean, stating its assumptions and mathematics in a transparent way, and I like the condition derived in Lemma 1 (despite being not tight in many cases). I found the heatmaps maybe not the optimal choice to plot the boundaries since the boundary is essentially a line in all of them (except maybe in Figure 7 in the appendix where the boundary seems less well defined). My concern is that the paper - at least to the extent that I understood it - doesn't provide a significant extension over existing knowledge in the field and the selected experiments although well aligned with previous work, don't seem to explore all interesting cases (or at least are not discussed sufficiently). While I may have misunerstood some parts, I think that disentagling the impact of cooperative/competitive underlying game on convergence and doing more extensive experiments in games not covered by (the limiting) assumption 1, would strengthen the paper. In general, while extending the analysis to other models of random graphs - unless they provide substantially different motivations and challenges - as mentioned in the conclusions may be interesting, it seems to me that going deeper into the present theoretical results may be more interesting - the breadth could be covered with more experiments.

---

> ### Author Response · Authors · 2025-11-23
>
> We thank the reviewer for carefully reading our manuscript.
>
> Regarding the reviewer's question on Assumption 1, we agree that it is a simplification. For this reason,
> we go beyond this setting in our experiments through the Conflict Network Game in which Assumption 1 is
> violated. We find empirical evidence that our predictions still hold (c.f. Figure 2 (right)), which
> we hope to explore in future work. Our goal in this work is focused on understanding the role of
> the network in multi-agent learning dynamics, rather than to consider the effect of cooperation and
> competition. Other works have considered the
> specific influence of cooperation and competition via decompositions of the payoff matrix (e.g.
> Hwang and Rey-Bellet (2011) Decompositions of two player games: potential, zero-sum and stable
> games). However, the application of such techniques in the study of learning dynamics is an open and
> interesting research question. We discuss our setting in more detail in the paragraph starting on line 242.
>
> Whilst it is known that Q-Learning converges for high exploration rates, we point out in line 252
> that it is desirable for Q-Learning to converge with lower exploration rates, as the resulting
> equilibrium solution is closer to a Nash Equilibrium (c.f. McKelvey and Palfrey (2005): Quantal
> Response Equilibria for Normal Form Games). Since it is also known that convergence \emph{does not}
> occur for low exploration rates, it remains to ask how low the exploration rate can be to guarantee
> convergence, and what factors in the game influence this bound. The contribution of our work is to determine the contribution of the network structure and provide probabilistic bounds. Importantly, the new results show that once we control for the expected degree, the threshold for chaotic behaviour increases from linear to exponential in N. In other words, with controlled expected degrees, chaotic dynamics can only emerge at exponentially large network sizes, far beyond the regimes considered in earlier
> analyses.

---

> > ### Author Response · Authors · 2025-11-23
> >
> > Response To Questions:
> >
> > Regarding the relation to Hussain et al. (2024): in that work, the authors also discuss the
> > influence of the network structure on learning stability, but they do not provide a controllable
> > quantity which may be used by practitioners to control the stability of the learning algorithm.
> > Rather, their results are framed in terms of adjacency matrix norms. In many real-world settings,
> > a practitioner cannot directly set, e.g. the two-norm, of the adjacency matrix for their network,
> > but can only do so implicitly by increasing or decreasing the probability of edge connections.
> > Consider, for example, a robotic swarm in which agents can communicate with neighbours within a
> > $x$-m radius. By increasing, or decreasing $x$, a practitioner varies the probability of connections
> > between agents. Our results provide an explicit link between the probability of edges and
> > stability of learning, which is only possible by transferring to the study of random networks. To
> > the best of our knowledge, such an analysis has not been conducted by other works on multi-agent
> > learning dynamics.
> >
> > Our random-network analysis also allows for the study of certain network classes, such as the Stochastic
> > Block Model, which is a well-studied model of distributed systems that contain sub-communities. An
> > example of such a system is a robot team, in which agents communicate with those within their teams
> > more often than those outside their team. With the tools of random networks, we may understand how
> > increasing or decreasing inter-, or intra- community edge connections will affect the resulting
> > convergence of the learning algorithm. In the robot-team example, a practitioner may not have control
> > over inter-community connectivity (often specific 'leader' agents are assigned for inter-team communications)
> > but can control intra-community connectivity. An analysis such as ours allows for an understanding
> > of how controllable parameters influences learning stability, whereas a naive analysis of matrix-norms
> > cannot yield such practical insights.
> >
> > For the Sato game, the similarity index is often low (it is the value of $\epsilon_X + \epsilon_Y$
> > which in our experiments is 0.2). If $\epsilon_X + \epsilon_Y = 0$, we return to a zero-sum game
> > and, in fact, convergence occurs for any $T > 0$. However, as Sato (2002) showed, non-zero values
> > along the diagonal lead to non-convergent behaviour even in the two-player setting.
> >
> > The Sato and Shapley games are well-studied in works on evolutionary game theoretic analyses of
> > learning, as they are been shown to produce complex or chaotic dynamics with a number of learning
> > algorithms in two player settings (e.g. van Strien and Sparrow (2006): Fictitious Play in $3\times 3$
> > games: chaos and dithering behaviour). We therefore chose them as the representative examples to study
> > as the number of players increases. We chose the Conflict game to move beyond Assumption 1, as we
> > previously discussed.
> >
> > Regarding the contrast with Sanders et al. (2018), our work is a refinement of their findings rather than a
> > contradiction. In particular, Sanders et al. (2018) found that, as the number of players increases,
> > Q-Learning (although they refer to the algorithm as Experience Weighted Attraction) shows complex or
> > chaotic dynamics for larger exploration rates. The implication, in their words is that "... complex
> > non-equilibrium behaviour, exemplified by chaos, is the norm for complicated games with many players".
> > However, their work did not account for how agents interact with one another. Rather they assumed
> > that each player's payoff is directly affected by the action of all others in the environment. This
> > is often not the case in real-world settings, where agents can only interact through some
> > communication network, or with their nearest neighbours. As we show in our work, when we account for
> > the network, we find that the results of Sanders et al. (2018) only hold if the probability of edge
> > connections does not decrease as the number of players increases. Indeed, for the case that the
> > probability of edge connections $p=1$ (i.e. all agents interact with one another), we recover the
> > results of Sanders et al. (2018). However, if $p$ goes to $0$ faster than $1 / N$ (i.e. the network
> > remains sparse even with many agents), Q-Learning still converges with low exploration rates, i.e. exploration rates which are at most logarithmic in $N$, rather than show chaotic dynamics.

---

> > > ### Comment · Reviewer_ckUe · 2025-11-28
> > >
> > > Thank you for the rebuttal. I acknowledge your response and have no further questions at this stage. My main concern remains that the theoretical results are obtained under limiting assumptions that are not extensively lifted in the experimental sections with some effects resulting in non-tight bounds or unclear links between causes and effects. As such, the main takeaways, while plausible, may have limited robustness. The new additions are clear, and the distinction between discrete and continuous time versions of QLD are adequately discussed, so this is fine to me. Thus, I will retain my score for now, but I am more favourably inclined, will be happy to discuss the evaluation with the other reviewers.

---

> > > > ### Author Response · Authors · 2025-11-30
> > > >
> > > > Thank you for your response. To clarify, Assumption 1 (identical bimatrix games across edges) is required to separate the effect of the payoffs on the learning dynamic from those of the network, which is the focus of the work. Despite this, the setting of identical payoffs for each agent appears often in applications, such as load balancing (where each node aims to maximise task completion with access to the same resources as all other nodes) or the coordination of identical autonomous production units in a decentralised manufacturing system, where the objective functions (e.g., minimising cycle time or energy usage) are shared across all unit interactions. We do, however, agree that the analysis of more general payoff settings is of interest to the community. We consider our work to be an important first step that introduces random network tools to the analysis of multi-agent online learning.
> > > >
> > > > Regarding our experiments, these were specifically chosen to isolate the effect of the network on stability. By holding the payoffs constant and varying only network parameters, these experiments show a direct link between the network and stability of learning dynamics. Furthermore, the results for the Conflict Network Game, which does not rely on the identical payoffs assumption, provides compelling evidence that our theoretical results may hold outside of the assumptions required to derive them. This robustness is a key contribution that we believe motivates future studies in this direction.

---

### Official Review · Reviewer_goju · 2025-11-01

**Soundness:** 3
**Presentation:** 3
**Contribution:** 2
**Rating:** 6
**Confidence:** 3

**Summary:**

This paper investigates the convergence properties of multi-agent Q-learning dynamics in network polymatrix games. The central contribution is to establish a theoretical link between the convergence of the learning dynamics and the structure of the underlying interaction network, particularly for large-scale systems. The authors focus on networks generated from random graph models (Erdős-Rényi and Stochastic Block models) and derive sufficient conditions for convergence to a unique Quantal Response Equilibrium (QRE). The key result is that the minimum exploration rate required for guaranteed convergence scales with the expected node degree of the network. This implies that convergence can be reliably achieved even in systems with a very large number of agents, provided the network is sufficiently sparse. The theoretical findings are validated through a series of numerical simulations that demonstrate this trade-off between network density, exploration rate, and the number of agents.

**Strengths:**

*   The paper tackles the challenge of ensuring convergence in multi-agent learning as the number of agents grows, a key barrier to deploying learning agents in real multi-agent scenarios.
*   It provides a clear and theoretically grounded relationship between network sparsity (expected degree) and the convergence of Q-learning.
*   To control network connectivity as a practical lever for ensuring stability, could be useful for practitioners and multi-agent systems.
*   The theoretical claims are backed by suitable experiments that clearly illustrate the predicted effects across different games and network models.

**Weaknesses:**

*   The main theoretical results (Theorems 1 and 2) depend on Assumption 1, which implies that every edge in the network represents the same bimatrix game. This is a strong simplification, and the homogeneous graph structure also limits applicability.
*   The analysis is performed on the continuous-time Q-learning dynamics (QLD), which is an idealized model. The paper does not discuss the potential gap between these results and the behavior of the discrete-time Q-learning algorithm that would be implemented in practice.
*   The convergence guarantees are asymptotic. While the experiments confirm the trend for finite N, a discussion on how tight the bounds are for practical, non-asymptotic cases could be useful.

**Questions:**

1.  Your theoretical results are derived under the strong Assumption 1 (homogeneous payoffs). Could you comment on the primary theoretical hurdles to relaxing this assumption? And how well would the methods work for general graphs from datasets?
2.  The analysis is based on the continuous-time QLD model. How do you expect these guarantees to translate to the discrete-time Q-learning algorithm?
3.  Theorems 1 and 2 provide a lower bound on the exploration rate for convergence. How does this theoretical bound compare to the empirical convergence boundary you found in your experiments (e.g., in Figure 2)?

---

> ### Author Response · Authors · 2025-11-23
>
> We thank the reviewer for reading our work in depth and for providing areas for clarification.
>
>
> Regarding the reviewer's question on Assumption 1, we agree that it is a simplification. For this reason,
> we go beyond this setting in our experiments through the Conflict Network Game in which Assumption 1 is
> violated. We find empirical evidence that our predictions still hold (c.f. Figure 2 (right)), which
> we hope to explore in future work. We use this assumption to disentangle the effects of the payoff
> matrices from the network when analysing stability. Note that the latter is our main focus in this
> work, as previous works have largely neglected the role of the network in stability analysis. To
> yield more results outside of Assumption 1, one would have to find an alternative route to
> disentangling the payoffs from the network, as both appear in the formulation of the reward in line
> 148. It is likely that one would have to make another simplifying assumption which is domain
> specific, or place some upper bound on payoffs (this latter approach was taken, e.g. by Parise and Ozdaglar (2019)).
>
>
> Regarding results on the discrete-time system, we agree that this is desirable for practitioners.
> For this reason, in our experiments, we simulate the discrete-time
> Q-Learning algorithm for a fixed (4000) number of iterations. We discuss this change in our comment to
> all reviewers.
>
> We acknowledge that our previous results were asymptotic in N for fixed values of p, meaning that varying both values at the same time was not fully justified. To this end, we have amended our theoretical statements to present the finite-sample bounds that hold for any $p$ and $N$ jointly. We also discuss this change in our comment to all reviewers.

---

> > ### Comment · Reviewer_goju · 2025-11-26
> >
> > I thank the authors for their response. My concerns regarding the limitations have been answered by the response. The limited asymptotic results have been improved. The limitations of theory still remain significant, although authors give empirical evidence through simulations that their ideas make sense. Hence, I will keep my score.

---

### Official Review · Reviewer_28pC · 2025-11-01

**Soundness:** 3
**Presentation:** 3
**Contribution:** 2
**Rating:** 4
**Confidence:** 4

**Summary:**

This paper investigates the convergence performance of continuous-time Q-learning dynamics in network games, focusing on random graphs generated from the Erdős–Rényi model and stochastic block model. When learning the game, each agent independently executes Q-learning updates with softmax strategies. Given sufficient exploration (sufficiently high softmax temperature),  the authors show the asymptotic convergence of the Q-learning dynamics to the quantal response equilibrium—a solution concept that approaches the Nash equilibrium as the exploration rate vanishes—when both the algorithm running time and number of agents tend to infinity.
Experiments on three game models are provided, discussed, and compared with the theoretical results.

**Strengths:**

- The paper is well-written and the presentation is clear.
- The perspective of exploiting the graph structure for convergence analysis is novel and important.
- The proposed method (Q-learning) and solution concept (quantal response equilibrium) make sense for the problem.

**Weaknesses:**

### Theoretical Claims

The first concern is more on the presentation of the convergence results (Theorems 1 and 2). The results are asymptotic in terms of both the algorithm running time and the number of players $N$. However, as $N\to \infty$, $T_k \to \infty$, making the convergence point a trivial strategy that uniformly randomizes over the action space, as noted in the paper.

Nevertheless, non-asymptotic convergence results are more desirable. Could the authors comment on the feasibility of deriving non-asymptotic results for either the number of agents or the running time?

The asymptotic nature of the results also cast shadows on the validity of the solution concept QRE, as the results may only hold for large $N$, which corresponds to a high exploration rate and thus renders the QRE uninformative.

### Related Work

Hussain et al. (2024) appears to be a closely related work based on the authors' description and is not a concurrent work as it is heavily cited in the paper. However, the only difference between the two works discussed in the paper is that Hussain et al. (2024) "consider deterministic graphs with specific structure". This is a vague comparison and casts doubt on the significance of the work. The authors should provide a more detailed comparison of the two works to clarify the novelty of the current work, including but not limited to what graph structures are considered in Hussain et al. (2024), how are they fundamentally different from the random graphs considered in this paper, and what results are obtained in Hussain et al. (2024).

Some works on learning dynamic graphon games might be relevant to the paper, as learning unknown payoff functions is the key setting of the paper and extending to Markov games.

### Other

Only continuous-time Q-learning dynamics are considered. The authors may want to discuss the feasibility of extending the results to discrete-time Q-learning algorithms, which are used in practice.

Quadratic payoff with Assumption 1 is a bit restrictive. Moreover, the authors may want to discuss how Assumption 1 affects the players' action space. If all players share the same payoff matrices $(A,B)$, at least it implies all players share the same action space cardinality $|\mathcal{A}_{k}|$. What other implications may Assumption 1 have on the action space?

**Questions:**

Please see Weaknesses.

---

> ### Author Response · Authors · 2025-11-23
>
> We thank the reviewer for reading our work carefully and posing questions which will help us in the
> exposition of our theoretical results.
>
> We first address the concerns regarding the QRE, and its asymptote at the uniform distribution.
> The QRE itself is a well-established solution concept, particularly for modelling agents with
> bounded rationality (e.g. McKelvey and Palfrey (1995): Quantal Response
> Equilibria for Normal-Form Games), for its relation to Q-Learning (Leonardos et al (2024)), and as a
> means for computing Nash Equilibria (e.g. Gemp et al. (2021): Sample-based Approximation of
> Nash in Large Many-Player Games via Gradient Descent). In the case of Q-Learning, it is indeed
> undesirable that the agents should learn the uniform distribution, which is why we would like
> Q-Learning to converge with a low exploration rate T.
>
> Regarding the possibility of non-asymptotic results in the running-time, we agree that this is indeed a desirable
> direction for future research. For this reason, in our experiments, we simulate the discrete-time
> Q-Learning algorithm for a fixed (4000) number of iterations. We find empirically that the
> discrete-time system yields the behaviour predicted by the continuous-time system. We discuss this change in our comment to all reviewers. We are not aware of any works (specifically for Q-Learning in normal-form games) that have
> attempted to translate asymptotic results to finite-horizon, discrete-time settings, but we hope
> that future work will build on our results in this direction.
>
>
> Regarding the relation to Hussain et al. (2024): in that work, the authors also discuss the
> influence of the network structure on learning stability, but they do not provide a controllable
> quantity which may be used by practitioners to control the stability of the learning algorithm.
> Rather, their results are framed in terms of adjacency matrix norms. In many real-world settings,
> a practitioner cannot directly set, e.g. the two-norm, of the adjacency matrix for their network,
> but can only do so implicitly by increasing or decreasing the probability of edge connections.
> Consider, for example, a robotic swarm in which agents can communicate with neighbours within a
> $x$-m radius. By increasing, or decreasing $x$, a practitioner varies the probability of connections
> between agents. Our results provide an explicit link between the probability of edges and
> stability of learning, which is only possible by transferring to the study of random networks. To
> the best of our knowledge, such an analysis has not been conducted by other works on multi-agent
> learning dynamics.
>
> Our random-network analysis also allows for the study of certain network classes, such as the Stochastic
> Block Model, which is a well-studied model of distributed systems that contain sub-communities. An
> example of such a system is a robot team, in which agents communicate with those within their teams more often than those outside their team. With the tools of random networks, we may understand how increasing or decreasing inter-, or intra- community edge connections will affect the resulting convergence of the learning algorithm. In the robot-team example, a practitioner may not have control over inter-community connectivity (often specific 'leader' agents are assigned for inter-team communications) but can control intra-community connectivity. An analysis such as ours allows for an understanding of how controllable parameters influences learning stability, whereas a naive analysis of matrix-norms cannot yield such practical insights.
>
> Regarding Assumption 1, we acknowledge that it is restrictive. To address this, we extend our analysis in the Conflict Network Game, where Assumption 1 is violated. Empirical results show that our predictions hold even in this case (see Figure 2, right), and we plan to explore this further in future work.

---

### Official Review · Reviewer_C1Y7 · 2025-11-01

**Soundness:** 2
**Presentation:** 2
**Contribution:** 2
**Rating:** 4
**Confidence:** 4

**Summary:**

This paper analyzes the convergence of continuous-time Q-learning dynamics (QLD) in network polymatrix games where each pairwise interaction shares identical payoff structure. Under this symmetry assumption, the authors prove that if the exploration (temperature) rate T exceeds the product of the "intensity of identical interests" and the spectral radius of the adjacency matrix, then the game is strongly monotone and admits a unique quantal response equilibrium (QRE) to which QLD converges almost surely (Lemma 1). They further derive lower bounds on the exploration rate by establishing asymptotic upper bounds on the spectral radius for Erdos-Renyi and Stochastic Block Model networks, relying on Vu (2007) for the almost-sure spectral bounds.

**Strengths:**

- The paper draws an explicit analytical link between network connectivity and learning stability.
- The combination of graph spectral analysis with learning dynamics is relevant to multi-agent RL communities.

**Weaknesses:**

- Overstated scope in the abstract and introduction:
Q-learning primarily studied in the context of Markov decision processes or Markov games with discrete-time updates. The paper's limited focus on normal-form games and the continuous-time counterpart is not explicit in the abstract and introduction until lines 64-72. More importantly, the restrictive Assumption 1 is not clear in the abstract and introduction.
- Implicit information structure:
It is not clear whether agents know the model or can observe the others' strategies. For example, (1) involves r_ki(x_{-k}(t)). Then, how agent k can compute it without knowing r_ki and x_{-k}(t). Note that x_{-k}(t) is the probabilistic strategy rather than the action taken. Furthermore, does that update take place for all actions i. Then, how the agent can know the reward associated with actions not taken without knowing the model. If the agent knows the model, this should be explicit since the literature review criticizes existing works using model knowledge.
- Restrictive symmetry assumption:
The core Lemma 1 depends critically on Assumption 1 assuming identical bimatrix payoffs on all edges. This limits generality. Furthermore, under homogenous formulations, mean-field games (with continuum of agents) and mean-field-type games (with many agents) also have convergent learning dynamics. The statement at line 042-043 saying "...as the number of players grows, non-convergent behavior becomes the norm." is misleading in that regard.
- Presentation of sufficient vs. necessary conditions:
Statements like "higher exploration rates are necessary for convergence" at lines 242-243 are misleading. The results only prove sufficiency, and counterexamples could exist where QLD still guaranteed to converge under weaker conditions.
- Lack of rigorous in asymptotic bounds:
The paper cites spectral radius results from Vu (2007) to establish almost-sure upper bounds but does not quantify the non-asymptotic behavior for finite N. A clear statement would say that for any \epsilon > 0 there exists sufficiently large N such that if T \geq Bound(N) + \epsilon then the convergence of QLD is guaranteed.
- Ambiguity in the "p\rightarrow 0" statement at liens 304-305:
The claim "if p \rightarrow 0 faster than 1/N, QLD converges almost surely as N\rightarrow \infty" lacks clarity and mathematical rigor. The dominant term p(N-1) can remain bounded, yet we also have \epsilon \sqrt{N}. Therefore, it is not clear with what rate QLD can converge. Furthermore, such a statement could be misinterpreted as by making p go to zero we can ensure convergence while a change in p implies a different game.
- Limited literature review:
The literature review should discuss the seminal works: Shapley (1964) provided a counterexample of two-agent three-action game in which fictitious play does not necessarily converge to unique equilibrium. There is also famous Jordan's game in which fictitious play may not converge. Hart and Mas-Colell showed that for any uncoupled learning dynamics (not incorporating opponent payoffs), there exists a game in which they do not converge to equilibrium. More importantly, some citations are not provided in the references. For example, Leonardos et al (2024) is cited at lines 94 and 192. However, there is no such reference in the references. In the references, there are two papers from Leonardos yet both also were not cited within the main draft.
- Minor comments:
-- Line 158: T_1,\ldots, T_N > 0.
-- Line 165: "the QRE converges to the Nash Equilibrium" is not a rigorous statement. Do you mean the set of QRE converges to the set of Nash equilibrium?
-- Line 175: ... a discounted history ... is misleading since in the RL context, the discount refers to a geometric decay.
-- Line 184: ... denote ...
-- Line 295: Lemma 6 is in the supplementary material. This should be highlighted. Furthermore, an explicit proof of Theorem 1 and 2 would make the paper more complete. Stating these theorems first and then proving them using auxiliary lemmas would improve the accessibility of the paper.

**Questions:**

In the supplementary material:
- Lines 965-970: The statement: "We decompose each edge into a half-edge along which A is played and a half-edge along which B is played" is not clear. Can you explain this and the following adjacency matrix decomposition over an example? For example, if we only have two agents, then N = [O, -A; -B, O] and N+N^T = [O, -A-B^T; -B-A^T, O]. Then, what does G_{k\rightarrow l} and G_{l\rightarrow k} refer to?
- Theorem 1.4 in Vu (2007) also has a lower bound on \sigma^2. Why is it not included in Lemma 8?

---

> ### Author Response · Authors · 2025-11-23
>
> We thank the reviewer for their thoughtful comments and suggestions.
>
> Regarding Q-Learning, the reviewer correctly notes that it is typically studied in the context of Markov games within the reinforcement learning community. In contrast, our work focuses on the bandit setting, specifically normal-form games. This approach aligns with evolutionary game theory analyses of Q-Learning dynamics (e.g., Leonardos and Piliouras, 2022: Exploration-exploitation in multi-agent learning: Catastrophe theory meets game theory).
>
> The extra section also addresses the reviewer's second question. The Q-Learning dynamic is a continuous-time deterministic approximation of the underlying Q-Learning algorithm, where each agent receives only bandit feedback (payoff evaluations) and chooses actions randomly. The algorithm gives rise to a discrete-time stochastic process. The Q-Learning dynamic is an ODE models the \emph{expected} behaviour of agents under this algorithm. Further details are in Tuyls et al. (2005), and Appendix F includes a discussion and illustrations showing the agreement between the continuous- and discrete-time systems (Figure 5).
>
> Regarding the symmetry assumption (Assumption 1), it is required for our
> analysis to separate the effects of payoffs and the network, with the latter being our main focus—a contribution largely overlooked in prior work on learning stability. Importantly, our experiments go beyond this assumption, providing empirical evidence that our results hold more generally, as illustrated in the Conflict Network Game (Figure 2, right). We hope future work can extend our theoretical results to broader settings. Nevertheless, our study is the first to link the probability of edge connections in a network to the stability of multi-agent learning.
>
> The reviewer is correct that our results are sufficient rather than necessary, and we have updated
> our phrasing to reflect this. In this paragraph, we also present a case (pairwise zero-sum games) in which
> Q-Learning converges for all $T>0$ regardless of the number of players:
>
> "As an example, restricting to pairwise zero-sum games, where $\delta_I = 0$, ensures that (QLD) converges as long as $T_k > 0$ for all $k \in \mathcal{N}$, regardless of $\rho(G)$. Our approach, in contrast, focuses on controlling the spectral radius, $\rho(G)$, while treating $\delta_I$ to be fixed a-priori."
>
> Indeed, there are cases (such as monotone games) in which QLD will
> converge for all T>0. Regarding Sanders (2018), their work showed that, in many games it is indeed
> that higher exploration rates are required. Quoting from their own abstract: "... complex
> non-equilibrium behaviour, exemplified by chaos, is the norm for complicated games with many players".
> As the reviewers point out, counter-examples (exact potential games, monotone games) can be constructed
> where this is not true. However, we argue that these settings do not reflect most real-world
> multi-agent interactions and so a deeper analysis of all factors, aside from the payoff structure,
> is required.
>
> Regarding our asymptotic bounds, we agree that our arguments were asymptotic in $N$ while keeping $p$ fixed. In response, we have improved our theorem statements to present finite-sample bounds instead, so that they hold for any $p$ and $N$ jointly. We discuss this change in our comment to all reviewers.
>
> We thank the reviewer for pointing to the early literature on fictitious play and no-regret
> dynamics. As we are afforded an extra page during the revision period, we have included the suggested works
> in our literature review (line 97). The reference for Leonardos
> et al (2024) is on line 633 (Exploration-exploitation in multiagent competition: convergence with bounded rationality).
> The citation for Leonardos and Piliouras (2022) appears in the Appendix rather than the main paper.
>
> Regarding mean-field games, whilst it is the case that convergence results have been found, these
> largely require an additional assumption on the payoffs: monotonicity. An example can be found in
> Perolat et al. (2021): Scaling up Mean Field Games with Online Mirror Descent. As far as we are
> aware, it is still unclear whether online learning algorithms converge in general payoff settings in
> the mean-field setting.
>
> Regarding the reviewer's questions:
> 1. In your two player example, we may refer to $G_{1 \rightarrow 2} = [0, 1; 0, 0]$ and $G_{2 \rightarrow 1} = [0, 0; 1, 0]$.
>     We describe this setting with a more complete three-player example in Appendix D (due to space
>     constraints, we cannot put this in the main draft).
> 2. The lower bounds on sigma was included in line 1035 of Lemma 8 of the previous version. However, in our revised version we lift the dependency on Vu (2007)

---

### Author Response · Authors · 2025-11-23
**Response to all Reviewers (Summary of changes made in manuscript)**

We would like to first thank all reviewers for carefully reading our manuscript and
providing opportunities for us to improve our presentation. Here we outline the changes
that we have made to our manuscript based on the reviewers' suggestions. We respond to the reviewers' specific questions below.


**Finite player bounds**
Originally, our theorem statements were asymptotic with respect to the number of players $N$ while keeping $p$ fixed. Following reviewer feedback, we have revised these to present finite-sample bounds that hold for both $p$ and $N$ simultaneously. These updates can be found on page 6, with further clarification at the top of page 7 and a revised Appendix C. Importantly, these results highlight that, once we control for the expected degree, the threshold for chaotic behaviour transitions from linear to exponential in $N$. This means that chaotic dynamics can only arise at exponentially large network sizes, well beyond the scope of earlier analyses.

**Relation of the Continuous-Time Q-Learning Dynamics to the Q-Learning algorithm**

Our paper presents theoretical results on the last-iterate convergence of independent Q-Learning, modelled through continuous-time Q-Learning dynamics. This approach is commonly used in works applying evolutionary game theory to learning algorithms (e.g., Sanders et al., 2018; Leonardos et al., 2024; Ostrovski and van Strien, 2014). However, while the theoretical results are based on the continuous-time model, our experiments use the discrete-time independent Q-Learning algorithm. We have clarified this distinction in Section 4 with the following statement:

"We simulate the Q-Learning algorithm on network games, where the network is drawn from the ER and SB models. Note that we simulate the discrete-time algorithm described in Section 3, rather than integrating the continuous-time model (QLD). This is to understand how well the theoretical predictions align with the results of the discrete-time update, which is the algorithm that is applied in practice."

To better facilitate the discussion relating the continuous-time model and the Q-learning algorithm for readers who are more familiar with the discrete-time formulation, we have included a new appendix section (Appendix F) on the foundations of the Q-Learning dynamics. This largely follows the exposition of (Tuyls et al (2005): An Evolutionary Dynamical Analysis of
Multi-Agent Learning in Iterated Games). We also point to the full derivation of the model provided in (Leonardos et al (2024): Exploration-exploitation in multiagent competition: convergence with bounded rationality). In this section, we have also included Figures which show the agreement between the vector field generated by the continuous-time dynamic and trajectories simulated by the discrete time Q-Learning algorithm.

**Smaller changes**

- Amended our abstract in line 17 to explicitly state that we study normal-form games
- Included earlier works (such as Shapley 1964 and Hart and Mas-Collel (2003)) in our related literature on non-convergent learning dynamics

---

### Meta-Review · Area_Chair_vMbF · 2026-01-16

**Summary:**

The paper investigates the convergence properties of Q-learning dynamics in multi-agent systems in QRE solutions, within the context of network polymatrix games where interactions are determined by random graph models. A sufficient condition for convergence is derived based on the connectivity properties (convergence is driven by the expected node degree of the network rather than the total number of agents). To guarantee convergence, the key technical idea is to guarantee uniqueness of the QRE and asymptotic stability within the framework of random graphs.

**Reviewer Concerns:**

The main concerns of the reviewers were the technical novelty of the results, as it was pointed out by C1Y7. The AC also believes that the main result of the paper is effectively lifted from the deterministic case established by Hussain et al. and applying to random graphs is not novel (as the probabilistic bounds on the spectral radius on \rho(G) are standard results). Moreover, there were few concerns about the limited related work and the simplicity of the model (too restrictive) but the AC did not take them into consideration. The AC believes that there is some conceptual novelty on this work but it is not enough for the paper to get accepted in a top conference like ICLR.

**Reviewer Scores:**

The AC does not believe that the rebuttal could not adequately address the concerns of the reviewers for the paper to get higher scores and pass the bar for acceptance. It seems it is below the bar.

---

### Decision · Program_Chairs · 2026-01-26

Reject